# Blocking D2/D3 dopamine receptors in male participants increases volatility of beliefs when learning to trust others

Nace Mikus [1,2] ✉, Christoph Eisenegger[1,3,10], Christoph Mathys[2,4,5], Luke Clark [6,7], Ulrich Müller[3,8], Trevor W. Robbins [3], Claus Lamm [1,11] ✉ & Michael Naef[9] ✉

The ability to learn about other people is crucial for human social functioning. Dopamine has been proposed to regulate the precision of beliefs, but direct behavioural evidence of this is lacking. In this study, we investigate how a high dose of the D2/D3 dopamine receptor antagonist sulpiride impacts learning about other people's prosocial attitudes in a repeated Trust game. Using a Bayesian model of belief updating, we show that in a sample of 76 male participants sulpiride increases the volatility of beliefs, which leads to higher precision weights on prediction errors. This effect is driven by participants with genetically conferred higher dopamine availability (Taq1a polymorphism) and remains even after controlling for working memory performance. Higher precision weights are reflected in higher reciprocal behaviour in the repeated Trust game but not in single-round Trust games. Our data provide evidence that the D2 receptors are pivotal in regulating prediction error-driven belief updating in a social context.

Knowing whom to trust with our money, information, or health is central to our personal well-being[1]. The ability to form beliefs about other persons' attitudes from their actions is therefore pivotal for successfully navigating our social world. Inflexible beliefs, particularly about intentions of others, can lead to thoughts of persecution or even paranoid delusions–a hallmark symptom of psychotic disorders[2–4]. Understanding the neurocomputational substrates of social inference is therefore essential for informing pharmacological treatments of psychotic symptoms.

When learning whether to trust another person, we often do so by observing their behaviour across repeated interactions. How behaviours of others affect our overall beliefs about their trustworthiness largely depends on how certain we are about the attitudes that presumably drive others' actions[5]. For instance, if we believe someone will be hostile or friendly towards us with high certainty, any gesture from them will not much change our belief about them. On the other hand, that same gesture from someone whose intentions we are unsure of will likely strongly shift what we think about them. This process of belief updating under uncertainty has been formalised within the Bayesian Inference framework, where beliefs are represented as probability distributions and the degree to which new information affects the updating of beliefs is modulated by the precision (the inverse of uncertainty) of those beliefs[6]. As in similar computational frameworks[7], the belief update is proportional to the deviation of the prediction from the actual

[1]Department of Cognition, Emotion, and Methods in Psychology, Faculty of Psychology, University of Vienna, Vienna, Austria. [2]Interacting Minds Centre, Aarhus University, Aarhus, Denmark. [3]Behavioural and Clinical Neuroscience Institute and Department of Psychology, University of Cambridge, Cambridge, UK. [4]Translational Neuromodeling Unit (TNU), Institute for Biomedical Engineering, University of Zurich and ETH Zurich, Zurich, Switzerland. [5]Scuola Internazionale Superiore di Studi Avanzati (SISSA), Trieste, Italy. [6]Centre for Gambling Research at UBC, Department of Psychology, University of British, Columbia, Vancouver, BC, Canada. [7]Djavad Mowafaghian Centre for Brain Health, University of British Columbia, Vancouver, BC, Canada. [8]Adult Neurodevelopmental Services, Health & Community Services, Government of Jersey, St Helier, Jersey. [9]Department of Economics, University of Durham, Durham, UK. [10]Deceased: Christoph Eisenegger. [11]These authors jointly supervised this work: Claus Lamm, Michael Naef. ✉e-mail: nace.mikus@univie.ac.at; claus.lamm@univie.ac.at; michael.naef@durham.ac.uk

outcome, termed a prediction error (PE), weighted by the precision of prior beliefs. On top of this, new information also reduces uncertainty about the outcome. When prior beliefs are highly uncertain, the weight on the PE will be high and beliefs will be highly volatile. Conversely, if beliefs are held with high precision, this leads to a down-regulation of the influence of PE on learning and lowers belief volatility. Inflexibility in forming beliefs about others proportionally to their actions can result from high precision of prior beliefs about others' attitudes. Yet, the neurocomputational and neurochemical mechanisms regulating the uncertainty of beliefs are poorly understood. In this study, we examined the effects of the antipsychotic drug sulpiride, a D2/D3 dopamine receptor antagonist, on the uncertainty of beliefs about another person's trustworthiness.

Seminal studies in animals have established that mesolimbic dopaminergic circuits carry PE signals that drive belief updating in various contexts[8–10]. However, dopaminergic midbrain neurons have been shown to be involved in various probabilistic computations that go well beyond phasic signalling of surprising rewarding events. Dopamine responses scale with outcome variance[11,12] and reflect temporal and perceptual precision[13–15]. Several computational accounts of brain function suggest that uncertainty or precision coding is the main unifying feature of dopamine's involvement in belief updating[16–19]. Through encoding of uncertainty of beliefs about the world and what action to perform, dopamine receptors are thought to adjust the weights on PEs and control action selection[20,21]. But while there is evidence for the involvement of dopamine receptors in processing uncertainty in action selection[22–25], evidence for their causal role in regulating the uncertainty of social beliefs and adjusting weights on PEs is lacking.

Dopamine receptors within the corticostriatal circuitry are ideally positioned to regulate PE-related signal propagation and encode precision[26,27]. One possible neurobiological substrate of precision is proposed to be the post-synaptic gain of neuronal populations reporting PEs, where synaptic gain refers to the amplification or attenuation of the pre-synaptic signal on the post-synaptic cell[20,28]. Post-synaptic D1 and D2 type dopamine receptors in the striatum have complementary effects on synaptic gain[26,29]. D1-like receptors increase the excitability of post-synaptic neurons, whereas D2-type attenuate signal propagation and decrease synaptic gain[29]. A prediction from this is that dopamine binding to D1 receptors would promote PE propagation and increase belief updating. In contrast, D2 receptor stimulation would reduce post-synaptic responses and attenuate changes in beliefs, leading to belief rigidity[30,31]. In line with this reasoning, when learning about others, blocking D2 receptors should increase the volatility (or rate of change) of beliefs.

Although some studies indeed showed that blocking D2-type receptors enhanced learning from positive feedback[32], led to pronounced PE-related activity in the striatum[33], and enhanced performance[34,35], there is also evidence for attenuated PE coding and greater variability in choice selection[25,36,37]. The inconsistencies of these findings raise several important considerations. First, when alternative choices are available, it is often unclear whether increased switching between available choice options arises from drug effects on belief updating per se or from the effects on decision-making strategies (see for instance[38]). Second, D2 dopamine receptors have a higher affinity for dopamine[39] and doses of D2 antagonists commonly used in studies with healthy participants might not be sufficiently high to block the D2 receptor driven regulation of the PE signal[40]. Third, administration of compounds binding to dopamine receptors can have different and even opposing effects on learning and decision-making, depending on genetic variation in baseline dopamine function[23,41,42]. Fourth, integrating novel information with prior experiences also relies on working memory[43], which is also affected by drugs that target dopamine receptors[44]. And finally, beyond the methodological limitations of previous work, most studies with

dopamine receptor antagonists have examined learning about abstract stimulus-outcome associations using secondary rewards, which makes the translation to more complex social interactions questionable.

In light of these considerations, the present study administered a relatively high dose of the selective D2/D3 receptor antagonist sulpiride (800 mg) or placebo in a randomised double-blind parallel group design to 78 male participants, preselected based on their Taq1a polymorphism. The drug dose was chosen to maximise the blockade of postsynaptic dopamine D2 receptors while still being safe[45]. Most previous work used doses of 400 mg which leads to an occupation of ~30% of D2 receptors[46]. Using 800 mg leads to more than 60% occupancy and increases the likelihood of sufficiently blocking the effect of D2 receptors. Furthermore, as mentioned above, the effect of D2 antagonists often interacts with baseline variation on dopamine function[42,47]. Taq1a polymorphism is one of the most widely investigated genetic variations of the D2 receptor. Individuals with at least one A1 minor allele have been shown to have higher presynaptic dopamine availability[48] and reduced D2 receptor density in some subdivisions of the striatum[49,50]. Blocking D2 receptors might therefore have a stronger effect on belief updating in that genetic subgroup.

We investigated social learning by asking the participants to learn about other players' trustworthiness through a repeated Trust game (Fig. 1a). In the Trust game the investor may choose to transfer any portion of their monetary endowment to the trustee[51]. The transferred points are then multiplied by the experimenter before being passed on to the trustee. The trustee can then either reciprocate in a way that equalises the payoff of the two players or betray and keep everything. Participants in our study played 25 rounds of the Trust game as investors against two other players that were preprogramed to mostly equalise ("good trustee") or mostly betray ("bad trustee"). Importantly, we told the participants that the other players had given their answers weeks before the study day. Therefore, their decision to equalise or betray did not depend on the participant's investment. With this procedure, we increased the likelihood that their investments reflected the degree of uncertainty they had about the other player's response and were not confounded by strategic investment strategies, or exploratory action policies. By asking the participants to learn about a stable feature, we also ensure that participants' behaviour did not reflect differences in beliefs in the task volatility (the likelihood that the other person changed their mind), which might have obscured more basic processes related to forming beliefs about others.

The main goal of the study was to test the hypothesis that blocking D2-type receptors increases belief updates by reducing the precision of beliefs about others, whereby we also hypothesised that this effect would be more pronounced in participants with genetically conferred higher endogenous dopamine levels. The Results section of the paper is structured as follows: we first looked at how sulpiride affected investment updates and how this effect was moderated by the Taq1a genotype. We then examined how the updates related to the back-transfer of the trustee, by examining the effects of the drug and genotype on reciprocal behaviour. We then turned to computational modelling to determine how sulpiride affects the course of each participant's uncertainty around the other player's trustworthiness. To evaluate to what degree the effects of sulpiride were due to actions on working memory we included data on spatial working memory performance into our parameter estimation process. Finally, to control for effects of sulpiride on sensitivity to social feedback unrelated to learning, we surveyed data from two single-round social interaction tasks, targeting positive and negative reciprocal behaviour.

With this we show that sulpiride has a profound effect on how healthy males update their investment more from one trial to the next this effect is driven by increased uncertainty around the other player's actions. Results from both behavioural analysis and computational modelling support the claim that the drug effect on belief updating

was more pronounced in participants with higher endogenous striatal dopamine levels (indexed by the Taq1a polymorphism).

## Results

### D2/D3 receptor antagonism increases investment updates

We employed a Bayesian multi-level linear model predicting absolute change in investment from the previous trial, including variables for Treatment (sulpiride or placebo), Trial and their interaction as predictors (refer to supplementary material for outcomes of alternative models). Figure 1b shows that, following sulpiride administration, participants on average updated their investments more than participants in the placebo group ($b = 0.633$, 95% Credibility Interval (CrI) [0.117, 1.115], proportion of the posterior distribution of the regression coefficient below 0 being P($b < 0$) = 0.005), with an effect size d = 0.239 (95% CrI [0.045, 0.42]). The difference in investment updates was most apparent in the last trial of the task ($b = 0.863$, 95% CrI [0.289, 1.411], P($b < 0$) = 0.002, d = 0.325, 95% CrI [0.109, 0.531]) and we also found a small effect size on the Trial*Treatment interaction ($b = 0.457$, 95% CrI [−0.069, 0.99], P($b < 0$) = 0.047, d = 0.172, 95% CrI [−0.026, 0.373]). As participants learned about the trustees, changes of investments from one to the next trial reduced, and this decrease across time was less pronounced in the sulpiride group.

To examine whether the effects of the drug were moderated by the Taq1a polymorphism we ran another model including a variable for Taq1a-specific genotype and Trustee as predictors with the four-way interaction between the two new variables, Treatment and Trial, including a random intercept and slope for the Trustee (Supplementary Fig. 1a, Supplementary Table 4). Again, we found a main effect of treatment ($b = 0.595$, 95% CrI [0.112, 1.098], P($b < 0$) = 0.008), and a significant three-way interaction between Treatment, Genotype and Trial number ($b = 0.053$, 95% CrI [0.01, 0.098], P($b < 0$) = 0.007), but found no credible evidence of a two-way interaction Treatment × Genotype (b = −0.284, 95% CrI [−1.266, 0.708], P($b > 0$) = 0.287). These analyses suggest that on average sulpiride affected investment updates comparably across both genotype groups, but in contrast to the A2 homozygotes, the effect in the A1+ group was time dependent.

### D2/D3 receptor antagonism increases sensitivity to social feedback in the A1+ group in the repeated trust game

To further understand how investment updates related to back-transfer from the trustee, we defined reciprocal trials as trials where participants either increased investments (or repeated the maximal investment of 10 points) following positive feedback and decreased investments (or repeated an investment of 0 points) following a betrayal (Fig. 1c, for exact definition see Supplementary Note 2). We found that sulpiride led to a higher proportion of reciprocal trials ($b_{logodds} = 0.339$, 95% CrI [0.048, 0.661], P($b_{logodds} < 0$) = 0.012). This effect was significant in the A1+ group ($b_{logodds} = 0.469$, 95% CrI [0.052, 0.914], P($b_{logodds} < 0$) = 0.015) but we found no credible evidence for an effect in the A1- group ($b_{logodds} = 0.209$, 95% CrI [−0.212, 0.643], P($b_{logodds} < 0$) = 0.162); however, we also found no credible evidence that there was a difference of drug effects between the two genotype groups ($b_{logodds} = −0.263$, 95% CrI [−0.867, 0.329], P($b_{logodds} > 0$) = 0.186). Furthermore, we found some support for a dose response effect, whereby sulpiride serum levels in the blood correlated with reciprocal trials in the A1+ group (b = 0.185, 95% CrI [−0.04, 0.41], P(r < 0) = 0.05), but found no credible evidence for a correlation in the A1- group (Supplementary Table 1). Similar, albeit weaker, effects were found when we examined to what extent the signed investment update was dependent on the back-transfer and how this differed across the drug and genotype groups (Supplementary Fig. 1b).

### No credible evidence of an effect of D2/D3 receptor antagonism on average investment behaviour or overall performance

Next, we investigated whether this higher change of investments from one trial to the next is reflected in average investment patterns (Fig. 2a,

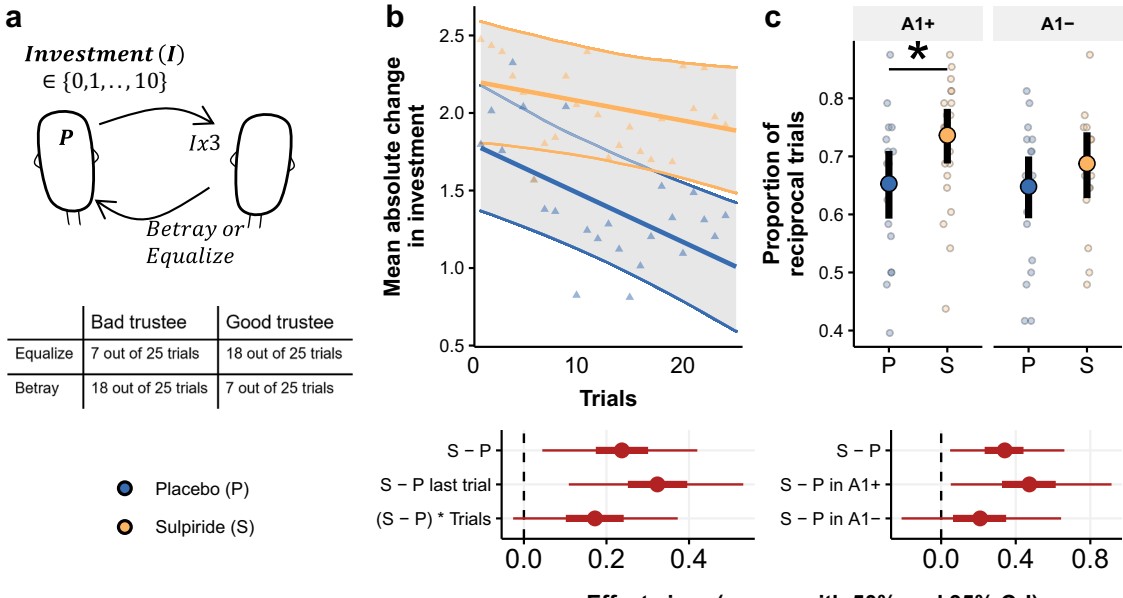

**Fig. 1 | Effects of sulpiride on investment updates in the repeated Trust Game.**
**a** The participants played 25 trials with two trustees. Each trial started with an endowment of 10 points to both players. On each trial they could invest any integer between 0 and 10. The trustee received a tripled amount of the investment and could decide to either equalise payoff or betray the other player and keep all the points for himself. The trustees were pre-programmed to be either "good" or "bad". **b** Mean and 95% CrI of absolute change of investment from one trial to the next for both treatment groups based on a Bayesian multilevel model, plotted over raw means for each treatment group (△), obtained over n = 76 participants, n = 38 in each drug group, and with 2 × 25 trials per participant. Corresponding effect sizes with means, 50% and 95% CrI, for the main effect of sulpiride (S-P), for the effect of sulpiride in the last trial and for the interaction of the drug with Trial variable. **c** Mean and 95% CrI of reciprocal trials (defined as trials where investment was increased following positive feedback, or decreased following negative feedback) based on a Bayesian logistic multilevel model, plotted over raw proportion of reciprocal trials with standard errors for each participant (sample sizes as in **b**). Effect sizes in log-odds shown for the main effect of sulpiride as well as the effect of sulpiride within each genotype group.

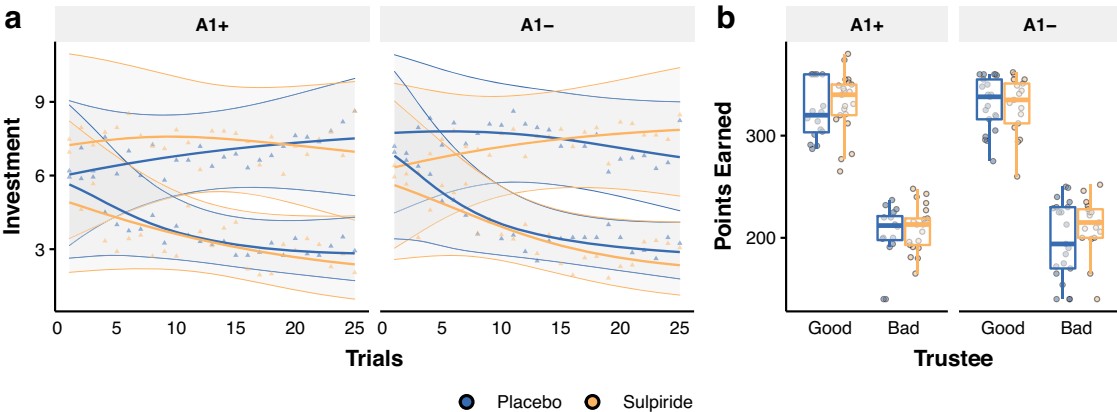

**Fig. 2 | Effects of sulpiride on average investments in the repeated Trust Game.**
**a** Average investment behaviour grouped for each trustee. Lines depict model predictions (means with 95% CrI as the shaded area), plotted over raw means for each drug group (△), obtained for 2 × 25 trials per participant, $n = 76$ participants (in A1+ group, $n = 17$ placebo, $n = 21$ sulpiride, and in A1- group $n = 21$ placebo and $n = 17$ sulpiride). We found no credible evidence for an effect of sulpiride in either genotype group on average investment behaviour regardless of the trustee. For statistics, refer to the main text and Supplementary Fig. 2. **b** Overall points earned in the task grouped for each trustee. Dots are average points earned for each participant. Boxplots with centre lines as medians, box bounds as 25th and 75th percentiles, and whiskers terminating at maxima/minima (a distance of 1.5 times the IQR away from the 25th and 75th percentiles). Sample sizes as in (**a**).

for more detailed plots see Supplementary Fig. 2). Using an ordinal logistic model predicting investments from Treatment, Genotype, Trustee and Trial variables, with a random slope for Trustee and Trial we found no credible evidence of a difference between sulpiride and placebo on average investment behaviour either in the A1+ group ($b_{good} = -0.011$, 95% CrI [−1.95, 1.796], $P(b_{good} > 0) = 0.494$, $b_{bad} = 0.089$, 95% CrI [−2.256, 2.47], $P(b_{bad} < 0) = 0.47$), nor in the A1- group ($b_{good} = -0.496$, 95% CrI [−2.353, 1.367], $P(b_{good} > 0) = 0.297$, $b_{bad} = 0.821$, 95% CrI [−1.508, 3.257], $P(b_{bad} < 0) = 0.244$). We also found no credible evidence of a difference in initial investments across the four drug and genotype groups (Supplementary Fig. 2). The overall initial investment was estimated to be, on average, 6.33 (95% CrI [3.129, 9.687]), suggesting that most participants expected a positive back-transfer initially. In line with this, the slope when playing against the good trustee was positive ($b = 0.086$, 95% CrI [−0.008, 0.187], P($b < 0$) = 0.036), but not as pronounced as the slope when playing against the bad trustee ($b = -0.153$, 95% CrI [−0.284, −0.027], P($b > 0$) = 0.009). While we found no credible evidence of a difference between slopes across the drug groups in the A1+ participants ($b_{good} = -0.058$, 95% CrI [−0.184, 0.065], $P(b_{good} > 0) = 0.179$, $b_{bad} = 0.044$, 95% CrI [−0.125, 0.21], $P(b_{bad} < 0) = 0.297$), we did observe an increase in the slope following sulpiride administration in the A1- group when playing against the bad trustee ($b_{bad} = 0.159$, 95% CrI [−0.007, 0.336], $P(b_{bad} < 0) = 0.03$) but not when playing against the good trustee ($b_{good} = 0.069$, 95% CrI [−0.06, 0.19], $P(b_{good} < 0) = 0.14$). Similarly, we found no credible evidence of a difference across drug and genotype groups regarding how many points they earned when playing against either trustee (Fig. 2b, Supplementary Table 13).

In summary, we found no credible effect of sulpiride on investing behaviour on average, but we do find some evidence in support of sulpiride increasing sensitivity to social feedback when learning about others. To determine whether and how this behavioural pattern relates to the uncertainty of participants' beliefs about the other persons' trustworthiness, we explicitly modelled the participants' trial-by-trial evolution of beliefs with a Bayesian belief model.

## Computational framework
The belief model uses a hierarchical Gaussian filter (HGF) to generate trial-wise sequences of participants' beliefs about the trustworthiness of two trustees as well as the uncertainty (or precision) surrounding those beliefs (Fig. 3[6,52]). We estimated a participant-specific parameter

$\omega$, called *belief volatility*, that describes how each participant's precision of beliefs evolved over time and consequently determined the relative rigidity (or malleability) of beliefs. More specifically, on each trial, we approximate the latent belief about the trustworthiness of the other player as a gaussian distribution with a specific mean and variance. Higher belief volatility $\omega$ implies higher variance (or lower precision) of trial-by-trial belief estimates. Importantly, the dynamic learning rate ($\psi_t$) on the PE is proportional to the expected variance or inversely proportional to the precision of beliefs and is therefore referred to as a "precision-weight". Low precision of prior beliefs leads to higher precision-weighted learning rates and stronger shifts in beliefs throughout the task (see two example belief trajectories with different $\omega$ values in Fig. 3b).

The beliefs about trustworthiness are mapped on to probability of positive or negative feedback with an inverse logistic function. Because D2 receptor activity is linked to choice uncertainty and action variability[22,25], we also included another parameter called choice precision parameter $\gamma$ that determined the non-linear mapping from beliefs to the investments. Higher choice precision implies an investment distribution centred around extremes (i.e. investing 0 and 10), and lower values imply a more dispersed investment distribution and more uncertainty or stochasticity in action selection. It thus mirrors the stochastic aspect of the inverse temperature parameter in the softmax equation often used in non-ordinal (e.g. binary) choice tasks. Finally, how beliefs about the probability of a positive back-transfer affect investment behaviour is determined through an ordinal logistic likelihood function. The degree to which inferred trustworthiness correlates with investments is determined by another parameter called the trustworthiness slope ($\eta$). Crucially, the computational parameters of the model represent distinct behavioural patterns and can be recovered reliably (Fig. 3c). To determine how noisy trials are represented in the model, we defined mistake trials as trials where participants either decreased their investment after a positive back-transfer or increased their investment after a betrayal (for exact definition see Supplementary Note 1). Importantly, we observed that belief volatility $\omega$ correlates with reciprocity ($r = 0.277$, $t = 2.476$, df = 74, $p = 0.016$, Fig. 3d) confirming that higher trial-by-trial uncertainty of beliefs lead to a higher chance of reciprocal behaviour. The log-transformed choice uncertainty parameter $\gamma$ correlates negatively with the proportion of mistake trials ($r = -0.592$, $t = -6.3254$, df = 74, $p < 0.001$, Fig. 3d) implying higher randomness in investment selection. We also

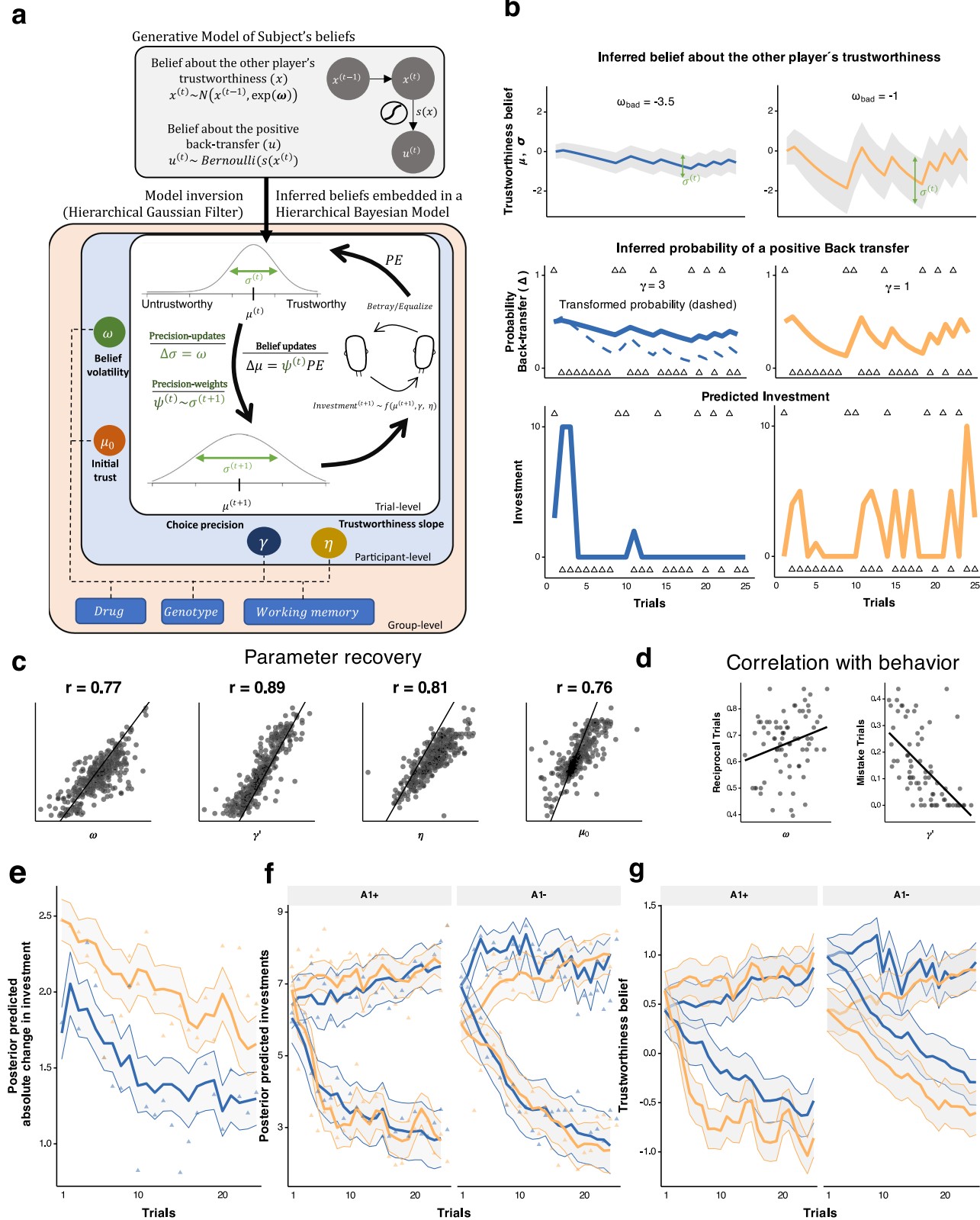

predicted data from the posterior distributions of parameters. We confirmed that the model captures the crucial aspects of behaviour (Fig. 3e, f) and plotted average beliefs about the other player's trustworthiness, grouped for each trustee. (Fig. 3g).

We compared this model to an HGF model without the $\gamma$ parameter and a Rescorla-Wagner (RW) model. The HGF model without $\gamma$ has been used previously when modelling both social[53] and non-

social[54] learning. In this model the non-linear mapping from beliefs to probabilities is modulated by a coupling parameter $\kappa$ (see Methods for details). The RW model is a simple Q-learning model with separate static learning rates for gains (positive outcomes) and for losses (negative outcomes). All models used the same ordinal-logistic likelihood function and were compared based on their trial-by-trial predictive accuracy through leave-one-out cross-validation information

**Fig. 3 | Computational modelling. a** We defined a generative model that describes the evolution of participants' beliefs about the other person's trustworthiness as a Gaussian random walk with the step size of $\omega$. The hierarchical Gaussian filter (HGF) inverts this model and provides trial-level estimations of participants' beliefs about the trustworthiness of others as Gaussian variables with mean $\mu^{(t)}$ and standard deviation $\sigma^{(t)}$. The evolution of $\sigma^{(t)}$ is determined by the belief volatility parameter $\omega$. The precision-weights $\psi^{(t)}$ are proportional to $\sigma^{(t)}$ and serve as dynamic learning rates when updating beliefs about the trustworthiness of the other player. We also estimate initial trustworthiness belief per participant ($\mu_0$). The ordinal logistic link function governs how beliefs about others' trustworthiness map to investments with two additional subject-level parameters: choice uncertainty ($\gamma$) and the slope ($\eta$). The parameter estimation is done through hierarchical Bayesian inference, where we estimate all individual and group-level parameters in one inferential step. **b** Two example belief trajectories portray the different behaviours that the model can capture, depicted as mean ($\mu^{(t)}$, line) and standard deviation of beliefs ($\sigma^{(t)}$, error band). The participants have different belief volatilities for the good ($\omega_{good}$) and the bad trustee ($\omega_{bad}$). Higher $\omega$ implies more uncertainty surrounding the trustworthiness beliefs ($\sigma^{(t)}$), which in turn leads to stronger belief shifts. **c** For each participant, we randomly draw parameters from their individual posterior distribution, simulate data, and re-estimate them five times. Relative high correlations indicate that the model parameters are well-defined. **d** The two main parameters of interest, belief volatility and choice uncertainty, correlate with distinct behavioural features, obtained for all participants ($n = 76$). **e,f** Posterior predictive for (**e**) absolute investment change from one trial to the next and (**f**) for the average investment behaviour. Plotted over raw means per trial per group and with standard deviations of predictions in the shaded area. **g** Lines depict average beliefs about the trustworthiness across participants for each trials, with error bands depicting average uncertainty around the investment ($\sigma$). All plots in (**e–g**) obtained with the following sample sizes: in A1+ group, $n = 17$ placebo, $n = 21$ sulpiride, and in A1- group $n = 21$ placebo and $n = 17$ sulpiride.

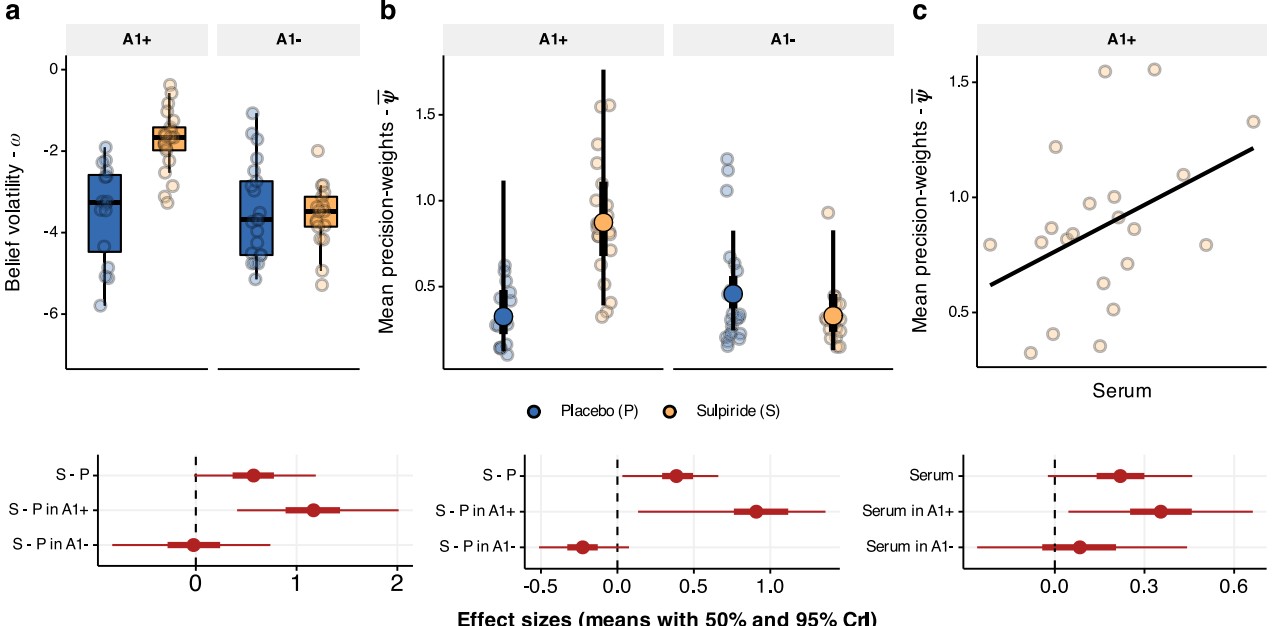

**Fig. 4 | Effects of sulpiride on belief volatility and precision weights. a** Belief volatility boxplots over individual means of posterior distributions. Boxplots with centre lines as medians, box bounds as 25th and 75th percentiles, and whiskers terminating at maxima/minima (a distance of 1.5 times the IQR away from the 25th and 75th percentiles). Sample sizes in A1+ group, $n = 17$ placebo, $n = 21$ sulpiride, and in A1- group $n = 21$ placebo and $n = 17$ sulpiride. Belief volatility is higher in the sulpiride group, and this effect is driven by the A1+ group (50% and 95% CrI of effect sizes below). **b** Precision-weights on PEs. Scattered points are meaned precision weights across all trials for each participant (sample sizes as in **a**). Overlayed group level medians with 50% and 95% CrI. The effect sizes were calculated from posterior distributions of differences in means across four groups. **c** Precision weights correlate with log transformed serum levels in the blood, plotted for participants in the A1+ group that received sulpiride ($n = 21$). The effect sizes with means (and 50% and 95% CrI) depict correlations between serum levels and median precision weights for the sulpiride group ($n = 38$), and then separately for the A1+ genotype group ($n = 21$) and for the A1- genotype group ($n = 17$).

criterion (LOOIC) and expected log predictive density (ELPD). We found that the HGF model with the choice precision parameter $\gamma$ outperforms both models (Supplementary Fig. 3a). We also compared the models across trials and across trustees with the LOOIC and by looking at the correlations of predicted investments with actual behaviour (Supplementary Fig. 3b, c). Interestingly, performance of both models varies similarly across trials with the HGF performing better across the whole task, particularly for investments against the good trustee.

**D2/D3 receptor antagonism increases belief volatility**
For parameter estimation, we embedded the HGF derived equations in a hierarchical Bayesian model which allowed us to estimate the drug and genotype effects on all computational parameters in one inferential step[55,56]. Through this analysis, we found a main effect of sulpiride on volatility of beliefs ($b = 0.831$, 95% CrI [0.115, 1.533], $P(b < 0) = 0.01$, d = 0.65, 95% CrI [0.088, 1.283], Fig. 4a), and an interaction effect of sulpiride with the genotype ($b = -1.506$, 95% CrI [−2.649, −0.411], $P(b > 0) = 0.004$, d = −1.175, 95% CrI [−2.238, −0.306]). In fact, the effect of sulpiride on belief volatility is driven by the A1+ allele carriers (b = 1.598, 95% CrI [0.727, 2.465], d = 1.25, 95% CrI [0.533, 2.119]) while we found no credible evidence of an effect in the A1- group (b = 0.076, 95% CrI [−0.874, 0.985], d = 0.06, 95% CrI [−0.683, 0.783]).

The key consequence of higher belief volatility is that it leads to lower precision of prior beliefs and therefore of predictions, which has a direct effect on the learning rates. Indeed, we founnd credible evidence that participants under sulpiride have higher average precision-weights (d = 0.452, 95% CrI [0.081, 0.704], $P(d < 0) = 0.008$, Fig. 4b), particularly in the A1+ group (d = 1.042, 95% CrI [0.225, 1.424], $P(d < 0) = 0.003$), but little credible evidence for an effect in the A2 homozygotes (d = −0.202, 95% CrI [−0.482, 0.103], $P(d > 0) = 0.089$) with a

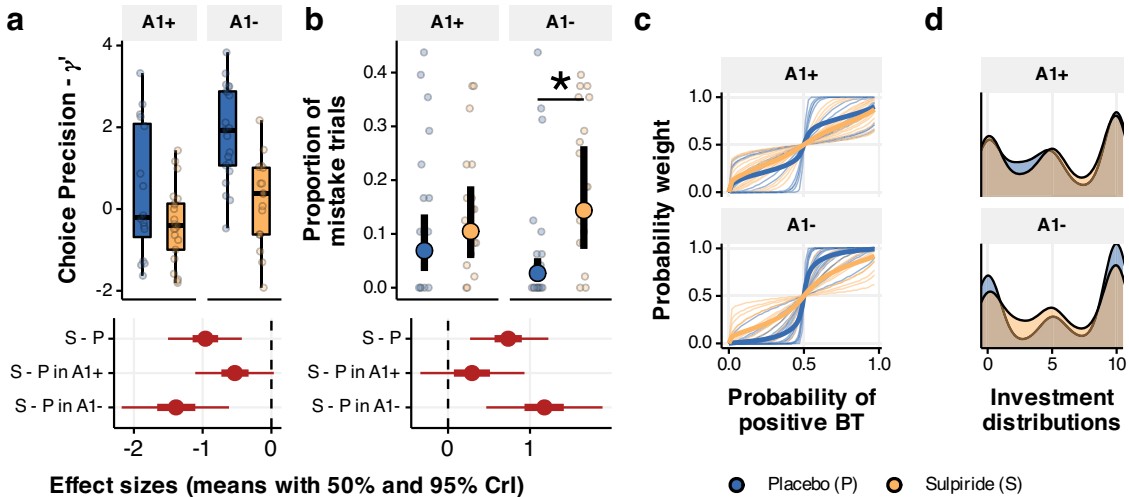

**Fig. 5 | Effect of sulpiride on choice precision. a** Effects of sulpiride on choice uncertainty, estimated in log – space (hence the prime). Dots are participant-level posterior means. The 95% and 50% CrI of effect sizes show a main effect of sulpiride, driven by the A1- group. Boxplots with centre lines as medians, box bounds as 25th and 75th percentiles, and whiskers terminating at maxima/minima (a distance of 1.5 times the IQR away from the 25th and 75th percentiles). Sample sizes in A1+ group, $n = 17$ placebo, $n = 21$ sulpiride, and in A1- group $n = 21$ placebo and $n = 17$ sulpiride.

**b** Mean proportion of mistake trials (with SEM), with samples sizes as in (**a**). Means and 95% quantiles of posterior distributions across the four groups are plotted based on a logistic regression model. Corresponding effect sizes below. **c, d** The choice uncertainty parameter determines the probability weight (**c**) and therefore the investment distribution (**d**). Higher values for the placebo group (the A1 group in particular) indicate more extreme investment choices and higher belief inflexibility.

significant interaction effect (d = 1.244, 95% CrI [0.335, 1.714], $P$(d < 0) = 0.001). Importantly, in the A1+ group, this effect of sulpiride on precision-weighting correlated with the degree of serum levels in the blood (b = 0.356, 95% CrI [0.045, 0.663], $P$(b < 0) = 0.013, Fig. 4c).

Looking at potential asymmetries when dealing with uncertainty around beliefs about trustworthy or untrustworthy partners, we find that, on average, the volatility of beliefs about the bad trustee were more volatile (d = 0.412, 95% CrI [0.031, 0.827], $P$(d < 0) = 0.018). When examining the drug effects, we observed that in the A1+ group, the difference in $\omega$ between placebo in sulpiride is apparent in interactions with both trustees ($d_{bad}$ = 1.62, 95% CrI [0.731, 2.644], $P$($d_{bad}$ < 0) < 0.001, $d_{good}$ = 0.689, 95% CrI [−0.089, 1.575], $P$($d_{good}$ < 0) = 0.042, but is higher for the bad trustee ($d_{good-bad}$ = −0.923, 95% CrI [−1.754, −0.121], $P$($d_{good-bad}$ > 0) = 0.01, Supplementary Fig. 4b, c). Interestingly, this analysis also showed that in the A1- group, there is a significant interaction of sulpiride and trustee effects ($d_{good-bad}$ = −1.453, 95% CrI [−2.529, −0.51], $P$($d_{good-bad}$ > 0) = 0.001, Supplementary Fig. 4b, c), whereby we find little credible evidence that the effects of sulpiride on belief volatility are higher for the bad trustee ($d_{bad}$ = 0.711, 95% CrI [−0.21, 1.669], $P$($d_{bad}$ < 0) = 0.066), and even negative for the good trustee ($d_{good}$ = −0.742, 95% CrI [−1.74, 0.091], $P$($d_{good}$ > 0) = 0.042). At this point, we also note that our model suggests that participants expected the trustee to reciprocate (d = 0.717, 95% CrI [0.406, 1.023], $P$(d < 0) = 0.001) with initial inferred probability of reciprocation being 0.67 (95% CrI [0.60, 0.73]). However, we found no credible evidence of a difference between treatment groups in initial beliefs ($\mu_0$) about the trustworthiness either overall (d = −0.166, 95% CrI [−0.697, 0.37], $P$(b > 0) = 0.272, Supplementary Fig. 4), in the A1+ group (d = 0.111, 95% CrI [−0.585, 0.802], $P$(d < 0) = 0.383), or in the A1- group (d = −0.441, 95% CrI [−1.164, 0.287], $P$(d > 0) = 0.112).

We then compared the results from the HGF model to those of the RW model (Supplementary Fig. 5). The evidence points in same direction, with some evidence for sulpiride leading to higher learning rates overall (d = 0.315, 95% CrI [−0.087, 0.729], $P$(d < 0) = 0.064, Supplementary Fig. 5a), an effect that the A1+ participants drove (d = 0.593, 95% CrI [0.03, 1.16], $P$(d < 0) = 0.02), with no credible evidence of an effect in the A1- participants (d = 0.037, 95% CrI [−0.524,

0.605], $P$(d < 0) = 0.453), and little credible evidence of a difference between the effect (d = −0.553, 95% CrI [−1.343, 0.235], $P$(d > 0) = 0.079). Further, the effect of the drug in the A1+ group was observed both when learning about positive outcomes (d = 0.697, 95% CrI [0.029, 1.369], $P$(d < 0) = 0.021, Supplementary Fig. 5b) as well as negative outcomes (d = 0.488, 95% CrI [−0.059, 1.044], $P$(d < 0) = 0.039, Supplementary Fig. 5c). However, the difference across the two types of learning rates was not as pronounced as in the HGF model.

## D2/D3 receptor antagonism increases choice uncertainty

In addition to the effect on belief volatility, sulpiride also increases choice uncertainty by decreasing the choice precision parameter $\gamma$ (b = −1.049, 95% CrI [−1.6, −0.502], $P$(b < 0) < 0.001, d = −0.979, 95% CrI [−1.535, −0.455], Fig. 5a), with smaller effects in the A1+ group (b = −0.646, 95% CrI [−1.272, −0.033], $P$(x > 0) = 0.02, d = −0.608, 95% CrI [−1.206, −0.031]) and more prominent effects in the A2 group (b = −1.44, 95% CrI [−2.261, −0.639], $P$(b < 0) < 0.001, d = −1.351, 95% CrI [−2.133, −0.601]). Since lower values of $\gamma$ correlated with higher proportion of mistake trials we examined how sulpiride affected the proportion of mistake trials and found that it on average increased the number of mistake trials ($b_{logodds}$ = 1.172, 95% CrI [0.443, 1.992], P($b_{logodds}$ < 0) < 0.001, Fig. 5b), an effect driven by the A1- group ($b_{logodds}$ = 1.876, 95% CrI [0.781, 3.032], $P$($b_{logodds}$ < 0) < 0.001) with no credible evidence for an effect in the A1+ group ($b_{logodds}$ = 0.468, 95% CrI [−0.535, 1.537], $P$($b_{logodds}$ < 0) = 0.184), and some evidence for an interaction effect ($b_{logodds}$ = 0.885, 95% CrI [−0.041, 1.857], $P$($b_{logodds}$ < 0) = 0.03). The effect of sulpiride on the proportion of mistake trials in the A1- group was proportional to the blood serum levels ($b_{logodds}$ = 0.607, 95% CrI [0.089, 1.142], $P$($b_{logodds}$ < 0) = 0.011) with no credible evidence of a correlation of the A1+ group ($b_{logodds}$ = −0.328, 95% CrI [−0.818, 0.143], $P$($b_{logodds}$ > 0) = 0.085). This parameter also determines the skew in the distribution of investments, whereby higher values make extreme investments more likely (Fig. 5c, d). From a perspective of an expected utility maximising agent, extreme investments are most optimal (Supplementary Note 2). Individuals with higher $\gamma$ therefore behave more as rational agents and take the uncertainty of the outcome less into consideration when choosing investments. Sulpiride also increased the $\eta$ parameter

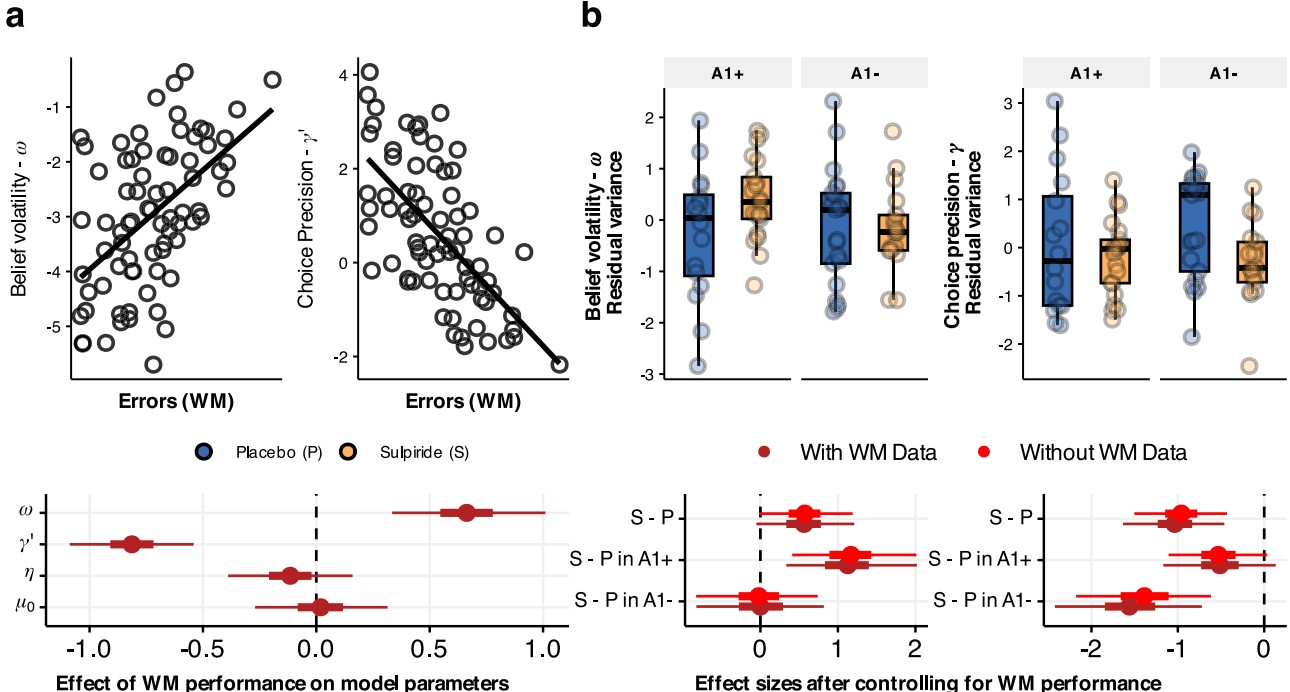

**Fig. 6 | Working memory performance and computational parameters. a** We reran the parameter estimation with a multilevel model that only included working memory data (number of errors) at the group level effect (and was agnostic about drug and genotype groups). Poorer performance in the spatial working memory task correlated positively with belief volatility ω and negatively with choice precision γ and did not affect noise or initial trustworthiness (obtained from sample size *n* = 75). Effect sizes depicted with means, 50% and 95% CrIs. **b** Residual variances after accounting for working memory data from the model that is agnostic about drug and genotype data, for parameters ω and γ. Boxplots with centre lines as medians, box bounds as 25th and 75th percentiles, and whiskers terminating at maxima/minima (a distance of 1.5 times the IQR away from the 25th and 75th percentiles), with the following samples: in A1+ group, *n* = 16 placebo, *n* = 21 sulpiride, and in A1− group *n* = 21 placebo and *n* = 17 sulpiride. In the second step, the parameters were estimated with working memory data and drug and genotype variables at the group level. The results of this analysis are shown below as effect sizes with means, 50% and 95% CrIs. The analysis is compared to that with the model that does not include working memory data.

(b = 1.459, 95% CrI [0.532, 2.42], *P*(b < 0) < 0.001, d = 0.941, 95% CrI [0.331, 1.58]), further advocating for the assertion that sulpiride increased the degree to which beliefs about trustworthiness influenced participants' investments. In sum, the overall results from the computational modelling suggest that sulpiride treatment led to higher choice uncertainty (lower choice precision), which was related to increased mistakes in the in the A1− group specifically. We found strong support for sulpiride increasing belief volatility and precision-weights on PEs, an effect that was driven by the A1+ group, whereas we found no credible evidence of an effect in the A1− group.

Effects of sulpiride on belief updating remain after accounting for working memory performance In the repeated Trust game, participants must remember the trustees' responses to previous trials. Higher choice stochasticity could therefore be due to poorer working memory. Furthermore, the inability to remember outcomes of past trials might increase the reliance on the previous trial and thereby cause increased learning rates and belief volatility. To determine to what degree our findings were influenced by the possible effects of sulpiride on working memory, we included data from a spatial working memory (WM) task performed in the same sample and previously published[57]. In the spatial WM task, participants uncover 'tokens' from sets of boxes, whereby they need to remember which boxes were previously searched and what the outcomes of those searches were (see Methods for details). As Naef et al. report[57], sulpiride had a detrimental effect on working memory performance, whereby participants in both genotype groups performed more errors (opened boxes previously already opened) in more challenging task trials (trials with 10 or 12 boxes).

In the present study, we first investigated whether the model parameters are influenced by WM performance. To do so, we re-estimated the hierarchical model that included only WM data at the

group level without the drug and genotype variables. As can be seen from Fig. 6a, the belief volatility parameter ω was associated with a higher number of errors in the WM task (d = 0.658, 95% CrI [0.334, 1.01], *P*(d < 0) < 0.001), and the Choice precision parameter γ negatively correlated with the number of errors (d = −0.811, 95% CrI [−1.087, −0.541], *P*(d > 0) < 0.001). This implies that poorer working memory performance is related to higher choice and belief uncertainty. Importantly, however, when plotting the residuals of the model parameters (unexplained variance after accounting for WM effects), the impact of sulpiride on belief volatility in the A1+ group can be seen still to be present (Fig. 6b). To obtain posterior distributions of drug and genotype effects after accounting for WM data, we re-estimated the parameters of the model this time including WM data as well as drug and genotype variables. We found that including WM information in the hierarchical model only slightly changed the inference about the effect of sulpiride on belief updating. The main effect of sulpiride on ω was now somewhat less certain with the 95% CrI including values below 0 (d = 0.56, 95% CrI [−0.052, 1.211], *P*(d < 0) = 0.036), but the effect in the A1+ group was still present (b = 0.852, 95% CrI [0.116, 1.61], *P*(b < 0) = 0.014, d = 0.694, 95% CrI [0.093, 1.369]). Similarly, posterior intervals of sulpiride effects on γ after including WM data were comparable to those without WM data, with the main effect remaining negative (d = −1.034, 95% CrI [−1.634, −0.461], *P*(d > 0) < 0.001), and the evidence for the effect is substantial in the A1− group (d = −1.562, 95% CrI [−2.423, −0.722], *P*(d > 0) < 0.001) and less so in the A1+ group (d = −0.512, 95% CrI [−1.168, 0.133], *P*(d > 0) = 0.056).

An important final step was to exclude the possibility that this increase in updating was due to increased sensitivity to social feedback in general, or due to decreased desire to maximise outcomes. For this, we turned to data from single-round social interaction

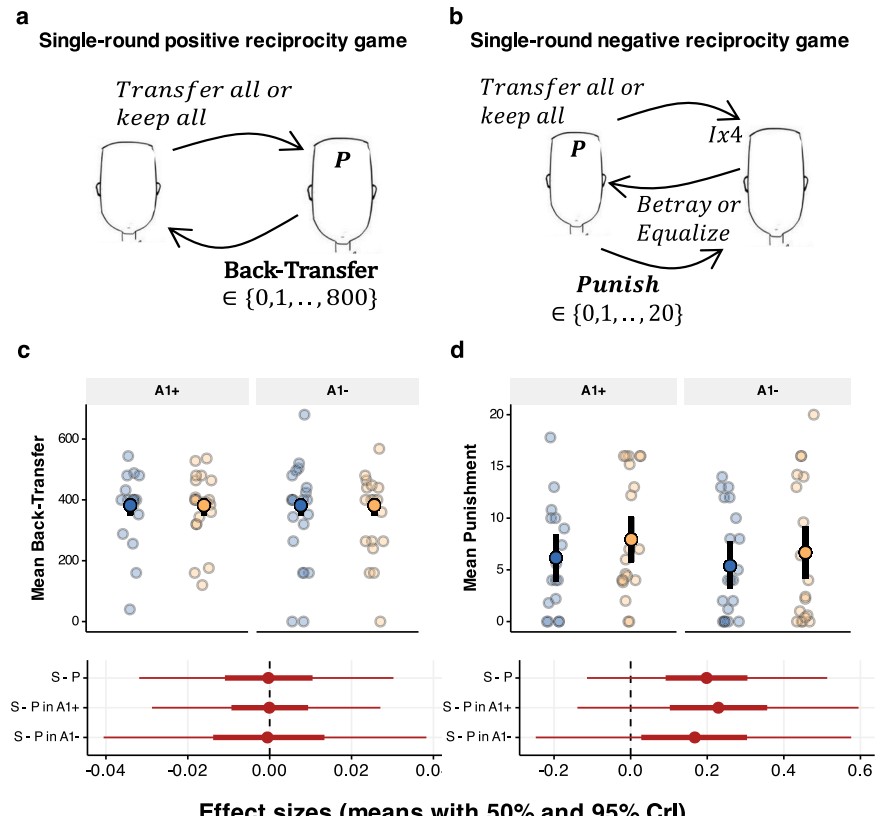

**Fig. 7 | Single-round reciprocity games. a** In the positive reciprocity task, participants played as trustees were paired with 7 other players. The investor in this version of the game received 800 points and could either keep everything or give everything to the trustee, who could then decide how to split the points. The investors were pre-programmed so that 5 out of 7 transferred everything to the trustee. **b** In the single-round negative reciprocity game, the participants played the investor. In the beginning of the round both players were given 10 points. The investor could then decide to transfer everything or nothing. The transferred investment got multiplied by a factor of four and the trustee could them decide to either equalise or betray. Crucially, after the choice of the trustee, both players received another 20 points and the investor could use his points to punish the trustee, with a factor of three. **c, d** Mean and 95% CrI of Back-transfer (**c**) and punishment (**d**) across the four groups, plotted over raw means (±SEM) per participant. Means, 50%, and 95% CrIs of effect sizes shown below, with the following sample sizes: in A1+ group, $n = 17$ placebo, $n = 21$ sulpiride, and in A1− group $n = 21$ placebo and $n = 17$ sulpiride.

games that measure learning-independent positive and negative reciprocity.

### No credible evidence for an effect of D2/D3 receptor antagonism on single-round reciprocal behaviour

In the single-round interaction games the participants played a slightly modified versions of the Trust game. In the positive reciprocity game, they played the trustee and could reward the investor for their decision (Fig. 7a). In the negative reciprocity game, they played as investor and could punish the trustee (Fig. 7b). We found no credible evidence of a difference between sulpiride and placebo, neither in the amount of reward (Back-transfer) in the positive reciprocity game ($b = −0.023$, 95% CrI [−6.605, 6.263], $d = 0.000$, 95% CrI [−0.032, 0.03], Fig. 7c) nor in punishing behaviour in the negative reciprocity game ($b = 1.552$, 95% CrI [−0.903, 3.98], $P(x < 0) = 0.106$, $d = 0.2$, 95% CrI [−0.114, 0.513], Fig. 7d). This implies that the drug effect on reciprocal behaviour in the Repeated Trust Game was not due to higher sensitivity to social-feedback, or to less rational behaviour.

## Discussion

Inferring attitudes of others is fundamental to our social functioning, but the neurocomputational mechanisms of the updating of beliefs about others are not well understood. We show that blocking D2/D3 dopamine receptors by sulpiride has a profound effect on how healthy participants process uncertainty in a social context. When playing as investors in the Repeated Trust Game, participants given a high dose

this D2/3 receptor antagonist changed their investment more from one trial to the next. Using a hierarchical Gaussian filter to explicitly model the evolution of participants' beliefs about the trustworthiness of the trustees, we show that sulpiride increased belief volatility. This implies that for the participants under sulpiride, the beliefs about the trustworthiness of others were held with less precision (i.e. with higher uncertainty), which in turn caused increased precision weights on PEs. This effect was more pronounced in participants with at least one minor A1 allele of the Taq1a polymorphism, associated with higher endogenous striatal dopamine levels. The increase in precision weights on PEs in that genetic subgroup scaled with the sulpiride serum levels in the blood. As a consequence, sulpiride led to higher reciprocal behaviour (increased investment after positive back-transfer and decreased investment after negative back-transfer), but only in the repeated Trust game, whereas we found no credible evidence for an effect in single-round interactions. The effect on the repeated Trust game was present even after controlling for working memory performance. Moreover, sulpiride decreased the value of the parameter of the model that codes for deterministic action selection policies ($\gamma$), implying higher uncertainty about investment selection. The effect was present in both genotype groups.

On the neurophysiological level, it has been proposed that precision is encoded through the post-synaptic gain (i.e. amplification or blunting of presynaptic neuronal input) of neurons that propagate PE signals[28]. Our results are in accordance with the idea that dopamine binding to D1 receptors of the medium spiny neurons in the striatum

increases the gain on PE signals, while binding to D2 receptors decreases gain through disinhibition of the so called indirect pathway[26,29]. Within this framework increased precision-weights following D2 antagonism can be explained by more dopamine being available to bind to D1-like receptors, a claim that is further substantiated by the observation that the effect of sulpiride was stronger in participants with genetically conferred higher presynaptic dopamine availability and lower D2 receptor density. These findings extend previous studies highlighting the role of dopamine receptors in coding precision or uncertainty in various contexts, such as perceptual and risk-based decision making[24,31,58]. In particular, previous work has shown that sulpiride decreased the perceived precision of temporal expectations[59]. In a task where participants were explicitly told about the variance of outcomes, they adapted their behaviour accordingly, which led to more optimal choice performance[60]. This behavioural pattern was accompanied by adaptive PE signals in the midbrain and the ventral striatum. Under 600 mg of sulpiride, both the PE scaling as well as the adaptive PE coding in the midbrain and partially in the striatum were reduced[61]. This suggests that D2 receptors likely play a general role in uncertainty coding across various task modalities and contexts.

Our findings that blocking D2/D3 receptors increases learning rates may seem to be at odds with previous work showing that D2/D3 antagonists reduced performance in other learning tasks and attenuated prediction error signals in the striatum[36,37] as well as with studies showing no effect of D2/D3 antagonism on learning rates[34,37], even when using similar computational frameworks[33,62]. It is thus important to note that the A1 is a minor allele of the Taq1a polymorphism, meaning that in most other studies participants were likely predominantly A2 homozygotes. We observed a more general effect of D2/D3 receptor antagonism on choice uncertainty that was more prominent in A2 homozygotes and was related to a higher number of mistake trials in that subgroup of participants, although the number of mistakes was not high enough to reduce investment on average. Furthermore, participants could invest on an ordinal 11-point scale, which allowed us to capture smaller belief shifts that might either be missed in learning tasks with categorical choice options or be attributed to a different choice selection policy. For example, the participants in our study also performed a standard probabilistic two-bandit task afterwards, where participants in the A1+ group under sulpiride compared to placebo continued to switch between choice options, which was explained by increased choice stochasticity, parametrised through the soft-max decision temperature[25]. Further, it is also plausible that the processing of uncertainty in a social context is different than in a nonsocial context. People might be inherently more motivated to reduce uncertainty about others, so that they can (for instance) classify them more definitely as being a friend or foe[5]. For example, in one study, stress increased the choice to gamble in a non-social context but decreased the likelihood to invest in one shot-Trust games[63]. Furthermore, patients with basolateral amygdala damage show markedly impaired belief updating in a repeated Trust game, but seem to have no trouble learning about non-social rewards through a task matched in difficulty and reward size[64]. It is therefore plausible that the results we found are specific to the social context and might not translate to learning about non-social cues.

One fundamental distinction that separates risky decision-making under a social compared to social conditions is an aversion to betrayal[65]. People are less risk-taking in social interactions and might be particularly sensitive to indications of untrustworthy interaction partners[66]. Using a similar model to ours, previous work has shown that belief volatility was higher when assessing (morally) bad agents[53], an effect that was present in our data as well, with participants having higher belief volatility when playing against the bad trustee. Although the drug effects on belief volatility were present across both trustees, the effects were stronger for the bad trustee. One reason for this could

be that because participants initially expected higher trustworthiness and higher rates of positive back transfers, there was more to learn when playing against the bad trustee and, therefore, more variance across investment behaviour. This asymmetric increase in sensitivity to negative outcomes would also be in line with the notion that D1 and D2 receptors in the striatum contribute to positive and negative outcome processing via the "Go" and "No-Go" pathways, respectively[32,67]. According to this circuit model, D2 antagonism mimics the dopaminergic 'dip' that occurs following negative reinforcement and therefore enhances learning to avoid action with a negative outcome. However, contrary to what we observed, this model also predicts that blocking post-synaptic D2 receptors should decrease positive prediction error propagation.

When interpreting our findings within the Go/No-Go framework, it should be noted that in the repeated Trust game in this study, participants have no agency over the valence of the outcome (positive or negative back-transfer), and investments are possible only on an ordinal scale. Similarly, the RW model we used in our study should also be interpreted with this in mind. Mirroring the effects in the HGF model, sulpiride increased learning rates in the RW model for both positive and negative outcomes. In multi-arm bandit tasks or Go/No-Go tasks where the reinforcement learning framework is often used to explain choice selection, the learning rate reflects the "Law of Effect" whereby actions that lead to positive (negative) outcomes are more (less) likely to get repeated[68]. A positive outcome following a specific investment choice in the repeated Trust game will lead to higher investment (if possible). A higher learning rate simply reflects the change in the expected response of the trustee. It is therefore related to the degree to which beliefs about trustworthiness change (on average across trials). Generally, the crucial distinction between RW and Bayesian models is that the latter assumes that agents consider the uncertainty of outcomes when updating beliefs. With this, Bayesian models such as the HGF or the Kalman filter can account for phenomena where non-Bayesian reinforcement learning models fail, such as latent inhibition and sensory preconditioning[18,69]. Given the relative increase in the overall and trial-by-trial predictive performance of the HGF model that includes choice uncertainty over the RW, our data support the notion that uncertainty about the outcomes and which actions to take affects choice behaviour in the repeated Trust game.

One important factor that could confound increased belief volatility and learning rates following sulpiride administration is working memory. Previous work has shown that individual differences in working memory capacity contribute to behavioural variability in reinforcement learning tasks[43,70] whereby decreased memory capacity might lead to a higher salience of more recent outcomes and therefore higher learning rates[71]. We also find support for this notion in our data, whereby poorer WM performance was strongly linked to higher belief volatility and higher choice uncertainty. However, despite sulpiride decreasing WM capacity in our cohort, including WM data in the model did not affect inference about sulpiride's effect on belief volatility, nor on choice stochasticity. This increase in choice uncertainty or stochasticity under sulpiride is therefore likely not due to failures in working memory capacity. Instead it could have been due to participants being less motivated to maximise outcomes, and therefore less likely to behave as a rational "homo economicus"[72]. Were that the case in our study, one would expect a different behavioural pattern in the single shot-Trust games. Participants under sulpiride should behave less as rational agents and therefore would be less likely to punish betrayals and reward trusting behaviour[73]. Note that D2 receptors generally do play a role in motivation. For example, optogenetic excitation and inhibition of D2 receptors in the ventral striatum of rats is reported to respectively increase and decrease motivation[74]. It is possible that the increased action variability resulted from reduced motivation, or increased noise in belief updating and not in choice selection per se[75]. What speaks against this interpretation is that the

overall performance in the task was not reduced following sulpiride administration for either of the genetic subgroups, suggesting that the investment selection under sulpiride was not random and instead reflected uncertainty about which investment to choose when interacting with the other player.

Indeed, variability in investment selection following sulpiride administration is well in line with what we know about the role of dopamine receptors in action selection. Stimulation of D2 receptors through endogenous dopamine leads to inhibition of the indirect (No-Go) pathway and increases the probability of repeating the same action. Accordingly, blockade of postsynaptic D2 receptors increases the probability of performing competing actions and therefore promotes randomness in action selection[76]. For example, in macaques, microinfusion of D2 (but not D1) receptor antagonists into the dorsal striatum led to increased choice stochasticity[77] and a similar pattern was observed in D2 receptor knockout mice[78]. In humans, a recent positron emission tomographic imaging study showed that D2 receptor availability in the striatum correlated with deterministic decision-making strategies represented either through decision temperature within reinforcement learning as well as with policy precision within active inference[22].

The key idea of active inference models is to extend the Bayesian generative models of beliefs about the states of the world, to include beliefs about preferred states, therefore casting both action and perception as an inference problem[79,80]. An active inference agent thus prefers actions that minimise the statistical distance (relative entropy) between the distributions of desired and predicted future states. The expected precision of a policy, in the context of our task, controls the confidence with which the participants selected a certain action, which can explain the more variable investment we observed in the sulpiride group. Within this framework, we can interpret the effects of sulpiride in our study as reflecting a more general role of D2 receptors in coding precision of both beliefs and action policies, thus extending previous theoretical and experimental work on the involvement of dopamine in modulating precision in predictive coding schemes[24,28,81].

Our findings might be particularly relevant for understanding the effects of antipsychotic medication in patients with psychosis, a disorder characterised by rigid beliefs of persecution, underlined by a profound lack of trust in others[82,83]. Previous studies with repeated Trust games showed that patients with psychosis have lower initial trust and find it hard to change their beliefs[3,84]. Neurocomputational accounts of delusions suggest that hyperactivity of D2 receptors in patients leads to increased precision beliefs that result in rigid convictions held with high confidence[20,85] and a recent paper shows that higher belief instability in patients with schizophrenia predicts responses to psychotherapeutic treatment[86]. This suggests that decreasing belief rigidity through D2 antagonism could be an essential contributor to the success of adjunct psychosocial treatment. However, there are profound differences between the effects of repeated use of antipsychotics in patients and acute D2 antagonism in healthy participants. For example, rodent studies show that although in healthy animals D2 receptor antagonism increases the activity of midbrain dopaminergic cells, this pattern is reversed in an established animal model of schizophrenia[87]. Furthermore, despite rapid receptor blockade of D2 receptor antagonists, the inhibition of excessive dopaminergic signalling proposed to underly the therapeutic effects develops only after weeks of treatment[88,89]. Our data also suggest that the therapeutic effect could be larger in patients that are A1 allele carriers of the Taq1a polymorphism. Yet, there is no evidence for this[90], despite higher dopamine synthesis being the most likely biomarker of psychotic symptoms[91,92] and a predictor of response to antipsychotic treatment[93]. Translating our findings to clinical practice will require more work with targeted patient populations.

Several important limitations should be kept in mind when considering the generalisability of the findings in this study. First, the sample was limited to male participants. This restriction was initially motivated by the notion that including female participants would require more than doubling the sample size due to increased variance of dopamine availability across the ovarian cycle. However, recent work shows no support for this[94]. Given that there are important sex differences in responses to antipsychotics, both in terms of efficacy and side-effect profiles[95], future work should prioritise studies in females. Second, despite our hypothesis-driven approach, the sample size of the genetic subgroups was small. The drug-gene interactions we report should be interpreted with this in mind. We note however that we did find a main effect of sulpiride on increased investment change and on belief volatility in a sample size comparable to other pharmacological studies with a between-subject design[96].

In conclusion, we show that blocking D2 dopamine receptors increases the flexibility of beliefs when learning about others. This finding importantly contributes to our understanding of how the brain infers the attitudes of other people. By mapping out the connection between alterations in the dopaminergic system with specific computational substrates this study not only contributes to the advancement of our knowledge of how the brain performs inference, but also to our understanding of when it fails to appropriately do so.

## Methods
### Participants
The study was performed in accordance with the Declaration of Helsinki and approved by the National Research Ethics Committee of Hertfordshire (11/EE/0111). Data were collected from 78 male participants, aged between 19 and 44 years (mean = 32.1), recruited from a large panel of participants, that were genotyped and screened for mental and physical health (Cambridge BioResource). Only participants with no history of neurological or psychiatric disorder were included in the study. Participants were stratified based on the Taq1A genotype into two groups: participants carrying at least one A1 allele (A1+), and A2 allele homozygotes (A1−).

### Procedure
After arrival participants underwent another psychiatric screening and an alcohol test to exclude alcohol consummation on the study day. After an assessment of general intelligence (National Adult Reading Test) participants signed an informed consent before they were administered a single oral dose of either 800 mg of sulpiride or placebo in a randomised, double-blind fashion. We used the parallel group design, because complex behavioural tasks (like the Trust game) have practice (repetition) effects that can confound the results of within group pharmacological experiments. Sulpiride maximal plasma concentration is expected to peak after 3 h, with a plasma half-life of about 12 h[97,98]. Before behavioural testing participants waited for 3 h in a quiet room, where they were allowed to read a newspaper. To monitor the effects of the pharmacological manipulation, blood pressure and heart rate and mood and drug effects were assessed prior to drug administration and after the 3 h waiting period. Similarly, blood samples to determine the serum levels were taken at both time points. After the blood draws (at around 3 h 20') the behavioural testing started with the social interaction tasks presented here, which included a repeated Trust game, and positive and negative reciprocity tasks, and were followed by a working memory task and an instrumental learning task, both published elsewhere[25,57]. Two participants were excluded from the analysis: one felt uncomfortable in the room, and one did not sufficiently understand the instructions of the social interaction tasks. This led to the following group distributions: 17 A1 allele carriers received placebo, and 21 received sulpiride, and 21 A2 homozygotes received placebo, and 17 received sulpiride. Participants were matched across the four groups for age, body mass index, general and verbal intelligence (Table 1, all $p > 0.30$). Participants received a monetary compensation of £50 plus the extra money earned in the behavioural tasks.

**Table 1 | Demographic information**

| Genotype | Treatment | N | IQ | sd | Verbal IQ | sd | BMI | sd |
|----------|-----------|---|-----|-----|-----------|-----|-----|-----|
| A1+ | Placebo | 17 | 120.2 | 5.333 | 120.335 | 5.913 | 26.098 | 3.125 |
| A1+ | Sulpiride | 21 | 120.21 | 7.334 | 120.355 | 8.133 | 26.595 | 5.74 |
| A1− | Placebo | 21 | 120.05 | 6.089 | 120.17 | 6.76 | 24.604 | 4.415 |
| A1− | Sulpiride | 17 | 117.659 | 7.494 | 117.518 | 8.318 | 25.064 | 5.393 |

*sd* standard deviation.

## Sulpiride serum concentration measurements

The level of serum sulpiride was determined by high-performance liquid chromatography. This method utilises fluorescence endpoint detection with prior solvent extraction. The excitation and emission wavelengths were 300 and 360 nm, respectively. Both intra- and inter-assay coefficients of variation (CVs) were 10% and the limit of detection was 5–10 ng/ml.

## Prolactin level assessment

The prolactin level was measured using a commercial immunoradio-metric assay (MP Biomedicals, Santa Ana, CA, USA), 3 h after capsule ingestion. Prolactin levels were expected to increase with blocking postsynaptic D2 receptors[97]. The intra- and inter-assay coefficients of variation were 4.2% and 8.2%, respectively, and the limit of detection was 0.5 ng ml−1[25]. We found that sulpiride administration significantly increased blood plasma prolactin levels ($\Delta = 33.1$ mg/ml, $p < 0.001$), and this increase was significantly higher ($p < 0.001$) than the changes in the placebo group ($\Delta = -0.91$ mg/ml, Mann Whitney test for differences $p < 0.001$). Data for three participants were excluded due to blood contamination.

## Side-effects and mood assessments

Side effects were assessed with a neurovegetative list[99], 3 h after drug intake. Mood was assessed with a visual analogue scale at baseline and 3 h after drug intake. Items in the visual analogue scales (VAS) were alert/drowsy, calm/excited, strong/feeble, muzzy/clear-headed, well coordinated/clumsy, lethargic/energetic, contented–discontented, troubled–tranquil, mentally slow/quick-witted, tense/relaxed, attentive/dreamy, incompetent/proficient, happy/sad, antagonistic/amicable, interested/bored and withdrawn/gregarious. The factors "alertness", "contentedness", and "calmness" were calculated from these items[100]. Data from one participant were excluded due to technical issues. We found no credible evidence of drug effects on mood, heart rate, blood pressure or self-reported side-effects (for details see Supplementary Material).

## Repeated trust game

In the Trust game[51] an investor (Player A) decides on how much money they want to transfer to the other player, called the trustee (Player B). The trustee receives the investment that is however tripled by the experimenter and decides on how to split the acquired sum. We used a multi-round version of the task[101], where the interchange between the investor and the trustee repeated across 25 trials. In the beginning of each trial both players were endowed with 10 points, to avoid investments motivated by inequality aversion. Each point converted to two pence at the end of the experiment. The participants could invest points on a scale from 0 to 10 and the trustee could respond in a binary fashion, by either equalising the payoff, or defecting by keeping all the points in the trial for themselves. Participants played as investors against two pre-programmed trustees: one defected in 7 out of 25 trials (the good agent) and the other defected in 18 out of 25 trials (the bad agent). The feedback was pseudorandomized separately for each participant and was interleaved whereby only two consecutive trials with the same trustee were allowed. To increase ecological validity, the

participants were led to believe that they play against two actual people that have already given their answers in advance several weeks before the testing, and that their decision will impact the payoff of these participants. All paradigms were programmed in Visual Basic.

## Positive and negative reciprocity games

In the positive reciprocity game, the two players need to distribute 800 points. First, player A is offered a distribution whereby they get 800 points and player B gets 0 points. They can decide to either keep all the points or delegate the decision on how to divide the points to player B. If the decision was to delegate, to player B can decide on any point distribution between the two players. Participants in our study played as player B sequentially against 7 different people playing as player A. The negative reciprocity game is like a Trust game in which defecting behaviour of the trustee can be punished by the investor. Both players are first endowed with 10 points. Player A then decides to either transfer his endowment (all the 10 points) all transfer nothing. The transfer of player A is quadrupled by the experimenter. Player B can then decide to either keep everything to themselves or to equalise the payoff. Following the decision of player B, both players get endowed with another 20 points and player A can spend each of these 20 points to penalise player B's outcome, whereby each penalty point of player A spent this way deducts three times as many points from player B's outcome. Participants in our study played as player A against 7 different people playing as player B. The actions of people playing player B were pre-programmed so that 5 out of 7 defected.

In both games the participants were told that the players have given their answers already days before the testing. Each point converts to 0.2 pence for the positive reciprocity game and 4 pence for the negative reciprocity game.

## Spatial working memory task

In the Spatial working memory task the participants were required to search through a spatial array of coloured squared boxes for a hidden 'token'[57] using a tablet. Participants have to touch the box to open it in order to reveal whether the token is in the box or not. When a token is found, the search starts again, whereby no token will be hidden in a box twice in the same trial. The performance measure is the number of search errors defined as errors committed when participants choose a box that has already had a token in that trial. There are 3, 4, 6, 8, 10 or 12 boxes in each trial, making the task progressively harder. The three boxes search were used as practice trials and a successful completion of them was a requirement for progressing onto the main test. The other difficulties each appeared three times. One subjects did not complete the working memory task and was removed from the analysis.

## Behavioural analysis

Behavioural analysis was done with Bayesian multilevel (generalised) linear regression[55], fitted with the brms package in R[102] through RStudio. All models were run with 4 chains, 3000 iterations each with 800 warm-up. The quality of chain convergence was inspected visually based on trace plots of main fixed effects, and a threshold on Gelman-Rubin $\hat{R}$ Statistic for each parameter was set to 1.01[103]. Throughout the behavioural analysis we z-scored the dependent variables (across the

**Table 2 | Prior distributions for the analysis of investment changes**

| Standard Deviations | $\sigma \sim HalfCauchy(0,2)$ |
|---|---|
| Regression Coefficients | $\beta \sim N(0,3)$ |
| Prior for the correlation matrix | $R \sim LKJcorr(2)$ |

whole group), coded the Treatment variable as 0 (placebo) and 1 (sulpiride), Back-transfer as 1 (equalise) and −1 (betray), Genotype as 0 (A1−) and 1 (A1 +) and centred the trustee variable (0.5 for good, and −0.5 for bad trustee). All random effects were modelled as a multivariate normal distribution, thereby evaluating the correlation between the effects as well as pooling information across the effects. Priors used are depicted in Table 2. The effect sizes where calculated by dividing the regression coefficients with the square root of summed variances of the residuals and of all random effects[104]. All models were redone also in the lme4 package[105] or nlme package[106], and the results of those models are reported in the supplementary material.

### Analysing investment behaviour
All model summary tables are in the supplementary material. The effect of sulpiride on absolute change from one trial to the next was evaluated with a model predicting effects on absolute change of sulpiride and trials with random intercepts for each participant. We also rerun the model including a participant-level slope for the trial and found that it does not affect inference about the main effect, but does increase the uncertainty around the interaction term. Next, the Genotype and Trustee as group level predictors were included as well as a random slope for the Trustee for each participant. Since the dependent variable is bounded at 0, the same analysis was done again with the dependent variable shifted by 1 and log transformed. This did not affect the conclusion of the model.

To analyse relative changes in investment the z-scored relative change from one trial to the next was predicted from the variable for Back-transfer (coded as −1 and 1), Treatment, Genotype, Trustee as well as their interactions, with a participant-level random intercept and slope for the Trustee.

The reciprocal and mistake trials were analysed with a multilevel logistic regression model including predictors Treatment, Genotype, Trustee, and their interaction, again with a random intercept and slope for the effects of the Trustee for each participant.

To analyse average investments, we used a multilevel ordinal-logistic regression model, with Treatment, Genotype, Trustee, Trial and their interaction, with a random intercept and slope for the effects of the Trustee for each participant.

Models analysing the single-round reciprocity tasks predicted Punishment (negative reciprocity) and Back transfer (positive reciprocity) from Treatment, Genotype and their interaction, including a random intercept per participant.

### Computational modelling
We first defined a generative model of the evolution of beliefs about the other players' trustworthiness as a Gaussian random walk. The belief volatility parameter $\omega$ describes the degree to which these beliefs can change from one trial to the next. We then used the Hierarchical Gaussian Filter (HGF) to invert this model[6].

### Generative model
The generative model describes the evolution of beliefs about the other person's trustworthiness as a Gaussian random walk with a step size of $\exp(\omega)$. In particular, at trial $t$ the belief on the other player's trustworthiness is defined as

$$x^{(t)} \sim N(x^{(t-1)}, e^{\omega}) \tag{1}$$

where $\omega$ is a participant level parameter. The mapping from the trustworthiness beliefs to the probability of a positive Back transfer ($BT$) occurs through a sigmoid transform $s(.)$. So, at trial $t$ we define:

$$BT^{(t)} = Bernoulli(s(x^{(t)}) \tag{2}$$

$$(x) = 1/(1 + e^x) \tag{3}$$

### Model inversion and update equations
To define the inferred participant level belief trajectories the generative model is inverted using Hierarchical Gaussian Filtering[6,52]. The HGF approximates full Bayesian inference using variational Bayes to derive at trial level update equations that resemble those of a Kalman filter[69,107]. In particular, the weights (learning rates) on the PEs are determined by the precision of prior beliefs as well as the uncertainty about the outcome. The HGF provides inferred posterior distributions of participants' belief trajectories as Gaussians through the mean $\mu^{(t)}$ and variance $\sigma^{(t)2}$ or its inverse, the precision $\pi^{(t)}$ in the update equations for both time series:

$$PE = BT^{(t)} - s(\kappa\mu^{(t)}), \tag{4}$$

$$\mu^{(t+1)} = \mu^{(t)} + \psi_t PE, \tag{5}$$

$$\pi^{(t+1)} = 1/\left(\frac{1}{\pi^{(t)}} + e^{\omega}\right), \tag{6}$$

$$\hat{\pi}^{(t)} = \pi^{(t+1)} + \left(s(\mu^{(t)})(1 - s(\mu^{(t)})\right), \tag{7}$$

$$\psi_t = 1/\hat{\pi}^{(t+1)}, \tag{8}$$

where $PE$ is the prediction error, $\psi_t$ is the precision weight (learning rate) that is determined by the expected precision of prediction ($\hat{\pi}^{(t)}$), and $\omega$ and $\kappa$ are free parameters in the model. This is a so-called recognition or perceptual model and describes our beliefs about the belief of the participants. To map the beliefs about the trustworthiness of the other person ($\mu^{(t)}$)on to the probability of feedback we used two versions of non-linear mapping from beliefs to probabilities:

$$\mu_p^{(t)} = pw(s(\kappa\mu^{(t)}), \gamma), \tag{9}$$

$$pw(x, \gamma) = \frac{x^{\gamma}}{x^{\gamma} + (1-x)^{\gamma}}, \tag{10}$$

where $pw(.)$ is a probability weighting function on the unit interval, with another free parameter $\gamma$ that determines the skew of the function. We estimated either $\gamma$ or $\kappa$ and fixed the other parameter.

We then mapped the probability of feedback to behaviour of the participant with a response model is defined through a likelihood function. Because investments occur on an ordinal scale we used the ordered logistic link function[108]:

$$P\left(I^{(t)} = k\right) \sim Ordered - Logit(\eta\mu_p^{(t)}C), \tag{11}$$

Where $C$ is a vector of intercepts and $\eta$ is the noise parameter that (similarly to the inverse temperature in the softmax equation) determines to what degree belief about the other person's trustworthiness determines investment behaviour. The ordered-logit estimates 10 intercepts, that determine the mapping from the linear term to the ordinal investments. In an ideal case, we would estimate all 10 intercepts for each subject, which was not feasible with our data. We

therefore estimate the 10 intercepts for all subjects, add one subject-level intercept in the linear term and assist the model in accounting for the various investment distributions with the non-linear mapping from beliefs to probabilities enabled either by $\kappa$ or $\gamma$. In the winning HGF model, $\kappa$ was fixed to 1 and $\gamma$ was estimated. In the second HGF model, $\gamma$ was fixed to 1 and $\kappa$ was estimated as a free parameter.

We also compared both HGF models to a simple Rescorla-Wagner model[109] with separate learning rates for positive and negative feedback. Given a learning rate $\alpha$, and an action value $Q$ of a chosen action on trial $t$, we defined the updating equation as:

$$Q^{(t+1)} = Q^{(t)} + \alpha PE, \tag{12}$$

$$PE = BT^{(t)} - Q^{(t)}, \tag{13}$$

while using the same likelihood function as for the HGF models. We estimated a different $\alpha$ for positive and negative outcomes.

## Parameter estimation
The model parameters were estimated in one hierarchical Bayesian model. This approach reduces overfitting[55], pools information across different levels (drug groups, and participants) and allows us to estimate both participant and group level parameters in one inferential step. Meaning, we estimate the effects of our drug manipulation on all relevant computational parameters in one model, while at the same time, leading to more stable parameter estimates[110]. Models were implemented in Stan[111] using R as the programming language and RStudio as the integrated development environment for R. Each candidate model with four independent chains and 3000 iterations (800 warm-up). Convergence of sampling chains was estimated through the Gelman-Rubin $\hat{R}$ statistic[103], whereby we considered $\hat{R}$ values smaller than or equal to 1.01 as acceptable.

The intercepts from the response model, $c_k, k = 1, \ldots, 10$, were estimated on the group level. This determined a general mapping from the probability to Investment. The participant level parameters ($\omega$, $\Delta\omega$, $\gamma$, $\eta$ and $\mu_0$) were modelled as a multivariate Gaussian distribution:

$$\begin{pmatrix} \omega \\ \Delta\omega \\ \gamma' \\ \eta \\ \mu_0 \end{pmatrix} \sim \mathrm{MVNormal}(\boldsymbol{\mu}, \boldsymbol{S}), \tag{14}$$

Where $S$ is the covariance matrix and $\mu$ is the vector of means. The $\Delta\omega$ parameter denotes the modelled difference in $\omega$ between the good and the bad Trustee. The matrix $S$ was factored into a diagonal matrix with standard deviations and the correlation matrix $R$[55,102]. The prime denotes the parameters in estimation space, whereby $\gamma$ was estimated in log space, due to it being lower bound by 0. The vector $\mu$ included all group level regression coefficients for the drug, genotype, and their interaction. The priors for group-level means for non-transformed parameters were weakly informative, for $\gamma$, estimated in log-space, the prior was $N(0,0.5)$, the prior for group-level standard deviations were more regularising, with $\sigma \sim Half Normal(0,0.2)$, and the prior for the correlation matrix was $R \sim LKJcorr(2)$. The prior for group level $\eta$ was set to something above 0, because chains that sampled from areas too close to 0 usually got stuck in that area. An overview of priors of parameters across the three models can be seen in Table 3.

To control to what degree working memory affects inference about the effects of drug and genotype we additionally estimated another model that included the number of search errors (z-scored) as a covariate on the group-level effect. To calculate the subject-level

**Table 3 | Priors for the three computational models**

|  | HGF with $\gamma$ | HGF with $\kappa$ | RW |
|---|---|---|---|
| Hyperpriors (random effects) | $\kappa - fixed$<br>$\mu_\omega \sim N(-2,1)$<br>$\mu_{\Delta\omega} \sim N(0,1)$<br>$\mu_\eta \sim N(10,1)$<br>$\mu_\gamma \sim N(0,0.5)$<br>$\mu_{\mu_0} \sim N(0,1)$<br>$\sigma \sim HN(0,0.2)$<br>$R \sim LKJcorr(2)$ | $\gamma - fixed$<br>$\mu_\omega \sim N(-2,1)$<br>$\mu_{\Delta\omega} \sim N(0,1)$<br>$\mu_\eta \sim N(10,1)$<br>$\mu_\kappa \sim N(0,1)$<br>$\mu_{\mu_0} \sim N(0,1)$<br>$\sigma \sim HN(0,0.2)$<br>$R \sim LKJcorr(2)$ | $\mu_\alpha \sim N(0,1)$<br>$\mu_{\Delta\alpha} \sim N(0,1)$<br>$\mu_\eta \sim N(10,1)$<br>$\mu_{\mu_0} \sim N(0,1)$<br>$\sigma \sim HN(0,0.2)$<br>$R \sim LKJcorr(2)$ |
| Priors of group level (fixed) effects | $\beta_* \sim N(0,1)$<br>$c_k \sim N(0,5)$ |  |  |

residuals after accounting for spatial working memory data, we ran another model that did not include drug data, meaning the only group level parameter affecting the group level means was the regression coefficient for te spatial working memory.

## Model validation and comparison
For parameter recovery 5 parameter sets were drawn from each participant's mean and standard deviation and used to simulate data. Simulated data were then estimated with the same model and the re-estimated parameters were correlated with the simulated ones. Further, posterior distributions of parameters were used to simulate data and check whether the crucial aspects of behaviour are captured by the model. A trial based Leave-One-Out Information Criterion (LOOIC) was used to compare the three models[112] using the loo package in R. The LOOIC approximate out-of-sample predictive accuracy of each trial, with lower LOOIC scores indicating better prediction accuracy out of sample. In addition, the performance of the models was compared with the with the "loo_compare" which return the difference (and the standard error) of each model to the best performing model in terms of expected predictive accuracy on a log scale (ELPD).

## Reporting summary
Further information on research design is available in the Nature Portfolio Reporting Summary linked to this article.

## Data availability
All data of the experiment is available online (https://doi.org/10.5281/zenodo.7779029).

## Code availability
The analysis scripts are available online (https://github.com/nacemikus/belief-volatility-da-trustgame.git).

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

## Acknowledgements

This paper is dedicated to the memory of Christoph Eisenegger. This research work was funded by the Vienna Science and Technology Fund (WWTF) with a grant (VRG13-007) awarded to Christoph Eisenegger and Claus Lamm. Christoph Eisenegger was also supported by the Swiss National Science Foundation (PA00P1_134135). We are grateful for the participation of all NIHR Cambridge BioResource (CBR) participants and thank the Cambridge BioResource staff for their help with participant recruitment. We also thank members of the Cambridge BioResource SAB and Management Committee for their support given to our study and the National Institute for Health Research Cambridge Biomedical Research Centre for funding. The paper has significantly improved from the constructive and insightful remarks of the three reviewers. Open access funding was provided by University of Vienna and Durham University.

## Author contributions

Study Design: C.E., M.N., U.M., L.C., and T.W.R.; Computational modelling: N.M., and C.M.; Software: M.N., and N.M.; Data analysis: N.M., and C.M.; Data Collection: C.E., M.N., and U.M.; Medical cover: U.M.; Resources: T.W.R., C.E. and C.L.; Data curation: C.E., N.M. and M.N.; Writing—Original draft: N.M.; Writing—reviewing and editing: all authors; Visualisation: N.M.; Supervision: M.N., T.W.R., C.M., and C.L.; Funding acquisition: C.E., and C.L.

## Competing interests

L.C. has received royalties from Cambridge Cognition Ltd. relating to neurocognitive testing. U.M. discloses consultancy for Janssen-Cilag, Lilly, Heptares and Shire, and educational funding from AstraZeneca, Bristol-Myers Squibb, Janssen-Cilag, Lilly, Lundbeck and Pharmacia-Upjohn. T.W.R. discloses consultancy with Cambridge Cognition Ltd and a research grant with Shionogi Inc. The remaining authors declare no competing interests.
