## [Peer Review File · Nature Communications]

Blocking D2/D3 dopamine receptors in male participants increases volatility of beliefs when learning to trust othersREVIEWER COMMENTS

Reviewer #1 (Remarks to the Author):

This is a well written paper addressing a clinically relevant and mechanistically informed, and informative issue, that is, the effects of a high dose of the antipsychotic D2 receptor antagonist sulpiride on the flexible updating of beliefs about trustworthiness. A repeated trust game with 25 trials is administered to 4 groups of subjects: one group of vulnerable A1+ carriers of the Taq1 allele, associated with altered D2 receptor density, on placebo; one A1+ group on sulpiride, one A1- group on placebo and one A1- group on sulpiride. Subjects invest money to play with a bad trustee, who defects their investment relatively often, and with a good trustee, who defects less often. The question is whether subjects learn to adapt their investments based on their experience with these trustees. Advanced mixed model analyses and hierarchical gaussian filter model-based analyses lead to the conclusion that sulpiride reduces the precision of beliefs about trustworthiness, thus increasing the weight on prediction errors. Subjects' investments vary more readily across trials after sulpiride than placebo, although they seem to converge on the same investment overall. This observation is relevant given the hypothesis that beliefs are overly rigid in psychosis, suggesting that antipsychotics might help render these beliefs more flexible.

Strengths of the paper include

- an intriguing question/timely topic that is clinically relevant,
- administration of a high dose of D2 receptor antagonist, ensuring sufficient receptor occupancy
- sophisticated Bayesian mixed model analyses of raw behaviour as well as model-based analyses
- rigorous execution of the analyses including model comparison, model validation with posterior predictive checks, simulations and parameter recovery;
- appropriate controls are in place, allowing the authors to conclude that effects on the repeated trust game do not reflect changes in e.g. the sensitivity to social feedback (as evidenced by absence of sulpiride effect on single-round social interaction tasks).

I have the following comments and questions to be addressed in a revision:

While the general rationale is clear and convincing, the basis for the directional prediction on precision weight on prediction errors in relation to the direct and indirect pathways in the introduction is unclear and hard to follow.

There was insufficient information in the paper about the alternative models against which the HGF model is compared, preempting evaluation of the implementation of e.g., the RW model and the associated conclusions.

This is relevant, because it was not clear why the effect does not reflect simply a higher learning rate, perhaps particularly for negative outcomes (a negative learning rate) and thus greater weight on recent versus remote (negative) outcomes. The paper would benefit from more extensive treatment of this alternative account. While a higher learning rate might reflect less precise or less rigid beliefs about trustworthiness, it might also have various other origins.

A (related) question is whether the pattern of (changes in) investments is better accounted for by a model that considers effects on working memory strategies. This is pertinent given sulpiride's established effects on working memory and the well-recognized finding that changes in RL in schizophrenia can reflect changes in working memory (e.g. Collins et al., 2014 and Collins et al., 2017). Might the drug have changed reliance on working memory versus reinforcement learning to elicit the increase in learning rate (see work by Collins and Frank)? To explore this alternative explanation, the authors could consider assessing changes in investment as a function of delay, but also recognize it in the discussion if applicable.

In this same context I wondered what is the optimal learning rate in this task, and was puzzled about

the data presented in supplementary figure 2, showing that subjects - on placebo - did not seem to learn to increase investments for the good trustee, while reducing investments across trials for the bad trustee. Isn't this a pattern one would want to see if subjects do the task as instructed?

While it is stated in the results section that average investments were not affected by drug or genotype, I did not see the statistics supporting this statement, and it was unclear whether this analysis was performed as a function of trial. I also suggest that beliefs for bad and good trustees are plotted as a function of trial, drug, genotype and trustee in the main text (much like Supp Fig 2 which I would think belongs to the main text too).

Critically, I would have thought that a hypothesis specifically about flexible belief updating is better addressed using a paradigm in which trustworthiness fluctuates across time at different frequencies (e.g. with some volatile task phases where the good trustee becomes a bad one and vice versa relatively frequently, and some stable task phases where contingencies are more stable). It is that type of setting that would allow a cleaner isolation of effects on learning rate per se and an effect on belief volatility.

Some example trial sequences are plotted in Figure 2b, but it is rather opaque what parameters are plotted on the y-axis. I suggest extending the legend considerably. And plotting the across-subject-average timeseries of investments for bad and good trustees (as suggested above, and in Supp Fig 2).

Finally, the observation at the end of the abstract and discussion section that the findings support the view that antipsychotics help to reduce the impact of distressing rigid beliefs seems to represent a jump-to-conclusions, particularly because the task does not involve reversals, or address the flexible updating of beliefs that are (now) inadequate. I note also that the paragraph preceding this last point on impairments in the formation of internal models, yet enhanced reliance on those internal models is a bit confusing.

Other comments:

Please specify in the paper whether the trial sequence is fixed pseudorandom, that is the same for both sessions and for all participants?

What was the delay between drug administration and administration of this specific task? What is time to peak for 800mg sulpiride.

Would one not have expected subjective effects of the high dose of sulpiride, e.g. on alertness and mood?

Sample sizes are somewhat modest, a point that might be emphasized particularly given the lack of an available preregistration

Reviewer #2 (Remarks to the Author):

In this study, the authors investigate how the administration of a selective D2/D3 dopamine receptor antagonist (sulpiride) impacts how we learn about the trustworthiness of others. Using a repeated trust game and a Bayesian belief model, the authors found that the administration of sulpiride led to (1) a higher uncertainty of the beliefs about the trustworthiness of others, but only in participants with at least one minor A1 allele of the Taq1a polymorphism (associated with higher endogenous dopamine levels); and (2) a higher uncertainty about investment selection, in both groups. Overall, the paper is very well written and meets the standards for publication. This work represents an important and relevant research topic in the field, the results are convincing and clearly presented. I find no errors

per se, and I only have a few minor remarks and clarification suggestions for the authors.

1) Page 2 § 2. In the introduction, the authors state "For instance, if we firmly believe someone is hostile towards us, a positive gesture coming from them will not much change our belief about them. On the other hand, that same gesture from someone whose intentions we are unsure of, will likely strongly shift what we think about them." I suggest that this should be phrased as an assumption since it is not supported by any reference and could even go against the findings from Siegel et al 2018, which were not cited by the authors but seem to be relevant for their study (in addition, the authors seem to find a similar pattern as described on page 6 § 4).

2) Page 5 § 3. The definition of parameters appears like a bullet list and leaves the reader to wonder why each of them is truly relevant in the model. It is specified later that each parameter describes a specific behavioral pattern, and this was supported by the model comparison, but it is not clear from this part of the manuscript.

3) Page 6 § 1. The acronym "HGF" needs to be defined here as well, in addition to the legend of Figure 2 and the Methods section.

4) Page 6 § 3. This is my main concern about the readability of the paper: I don't think the precision weights have been properly defined so far. Does it refer to the weight on PE or on prior beliefs? In which case, how does this relate to learning rates? The precision weights are such an important part of the results that they should be more clearly defined, as well as the consequences of them being higher or lower.

5) Page 8 § 3. Some parts of the discussion will benefit from a more cautious formulation or interpretation. For example, we know from several studies (e.g., FeldmanHall et al 2015 involving matched gambling and trust game tasks) that participants exhibit different behavioral patterns, including uncertainty valuation, in social vs. non-social tasks. Therefore, the results found in this social trust game might not be applicable to non-social reinforcement learning tasks.

6) Page 9 § 2 and related to point 5. The statement "suggesting that the investment selection under sulpiride was not random, but only less precise" is quite difficult to grasp and could be reformulated. It is unclear compared to the first paragraph of the discussion, where a variation of action selection parameter led to more or less stochastic choices.

7) I suggest the authors define the variables in Supplementary Note 1.

References

- FeldmanHall O, Raio CM, Kubota JT, Seiler MG, Phelps EA. The Effects of Social Context and Acute Stress on Decision Making Under Uncertainty. *Psychol Sci.* 2015 Dec;26(12):1918-26. doi: 10.1177/0956797615605807. Epub 2015 Nov 5. PMID: 26546080; PMCID: PMC4679610.
- Siegel JZ, Mathys C, Rutledge RB, Crockett MJ. Beliefs about bad people are volatile. *Nat Hum Behav.* 2018 Oct;2(10):750-756. doi: 10.1038/s41562-018-0425-1. Epub 2018 Sep 17. PMID: 31406285.

Reviewer #3 (Remarks to the Author):

Mikus and colleagues present a study that uses pharmacological antagonism of dopamine D2-like receptors to evaluate the role of dopamine in trusting behavior with a specific interest in belief updating. They apply a hierarchical gaussian filter to model choices made in the trust game and show that the D2-like receptor antagonist sulpiride increased choice uncertainty, the volatility of beliefs, and the extent to which this volatility influenced investments. The authors identified specific associations that depend on Taq1A genotype polymorphism, indicating that endogenous DA is an important moderator of general precision coding. I applaud the authors' for their careful consideration of alternative explanations and important checks in the form of modified versions of the task and comparison of their model to a simple learning model. The relation between dopamine and decisions in social contexts are greatly understudied, so this work is a welcomed contribution to the study of the neurobiology of decision making. I found the description of the modeling and methods to be clear. In general, I have few major concerns.

Major Comments:

1.) I am concerned about unintended consequences of a high dose of sulpiride. The Mehta et al., 2008 work cited, for instance, indicated that even 400 mg produces working memory deficits. Are these the same subjects reported in Naef et. al., 2017 also showing working memory impairment from 800 mg of sulpiride? Can the authors provide any assurance that such deficits could not account for the observed increased belief volatility or increase in choice uncertainty? Work by Anne Collins & Michael Frank, for instance, has shown that working memory is a contributor to reinforcement learning. Would such explanations affect interpretation of the current findings if the observed effects of sulpiride are working-memory related? In general, I am concerned whether working memory capacity may be more central to the mechanisms here than is currently discussed.

2.) The justification for the inclusion of only males in the study warrants mention in (1.) the methods and (2.) as a limitation should be mentioned in the discussion. It must also be clear in the abstract. While it is true that historically females have been excluded from some studies involving dopaminergic modulation over concerns of variation across the ovarian cycle, more recent evidence (I assumed published before the data presented here was collected) has failed to find differences in D2 receptor availability across the ovarian cycle (Petersen et al., 2021) and other groups have taken steps to ensure equal consideration (e.g. studying at specific phase of the ovarian cycle). One purpose of this kind of caveat is to ensure that future replication efforts and related studies do consider enrollment of female research subjects.

3.) The authors provide clinical relevance of their findings to psychosis treatment. One concern here is the extent to which comparisons can be made between mechanisms in healthy and clinical groups. For example, meta-analytic evidence indicates that the Taq1A polymorphism in schizophrenia does not affect responsiveness to antipsychotic pharmacotherapy (Zhang et al., 2010). This would seem at odds with the observation that some sulpiride-induced effects depend on Taq1A carrier status. Similar baseline differences in synthesis capacity have been observed in schizophrenia (Fusar-Poli & Meyer-Lindenberg, 2013; Howes et al., 2012). Recognition of these kinds of differences will be helpful when drawing inferences about psychopathology.

4.) There are a couple of places in the manuscript where an extra sentence or even a word or two could go a long way to aid interpretation of the results. For instance, "Importantly we observed that belief volatility ω correlates with reciprocity...and that the log transformed choice uncertainty parameter γ correlates negatively with the proportion of mistake trials...implying higher randomness in investment selection." Here there is a clear interpretation for the gamma but not omega parameter. Similarly, in the first paragraph of the section titled "Genotype-dependent effects of D2/D3 receptor antagonism on belief volatility and precision weights", there is no statement about the directionality of effects, just whether there are significant main or interaction effects on belief volatility. This contrasts the following paragraph where there is a clear statement ("participants under sulpiride have higher precision weights"). So, some clarity and consistency throughout the paper would go a long way.

5.) Can the authors provide more interpretation on the significance of belief volatility being higher for the bad trustee? While the interactions observed for this effect are weaker, they might hold interesting value for understanding the specific conditions under which precision of priors depends on negative outcomes or even the relationship between trustworthiness and belief shifts.

Minor Comments

1.) For what it is worth, there is additional work worth citing for the justification of using 800 mg of sulpiride. In clinical groups, at least, (Farde, 1988) show that in patients with schizophrenia, 800 mg (400 twice daily) corresponds to D2 occupancy between 68-82%.

2.) Related to the general point about clarifying effects/directionality, the authors state "At this point we also note that there were no differences in initial beliefs (μ_0) about the trustworthiness" – it is

unclear whether “differences” are referring to genotype or drug condition.

- 3.) Supplementary Figure 4 needs some kind of detailed caption like the rest of the figures.
- 4.) Potential typo: In the statement “sulpiride led to higher reciprocal behaviour (increased investment after negative back-transfer and decreased investment after positive back-transfer)” should it be increased after positive and decreased after negative?
- 5.) It is not clear what the y-axis labeling of top two plots in figure 2B are representing. The image itself is unclear but a parenthetical note or something of that nature in the caption could help.
- 6.) Were dependent variables z-scored within subjects or globally across all the data?
- 7.) Potential typo: In the “Parameter Estimation” section of the methods, it reads 3200 iterations but from the code and description in the “Behavioural Analysis” it should be 3000, I think?
- 8.) Typo in Fig 4a caption: “driven pa the A1-group.”

Overall, Mikus and colleagues present an impressive neurocomputational account of dopaminergic modulation of belief updating in a social context. This work will be of interest to social/cognitive psychologists, neuroscientists, and clinical investigators and provides an important contribution to the study of decision making.

References

- Farde, L. (1988). Central D2-Dopamine Receptor Occupancy in Schizophrenic Patients Treated With Antipsychotic Drugs. *Archives of General Psychiatry*, 45(1), 71. <https://doi.org/10.1001/archpsyc.1988.01800250087012>
- Fusar-Poli, P., & Meyer-Lindenberg, A. (2013). Striatal Presynaptic Dopamine in Schizophrenia, Part II: Meta-Analysis of [18F/11C]-DOPA PET Studies. *Schizophrenia Bulletin*, 39(1), 33–42. <https://doi.org/10.1093/schbul/sbr180>
- Howes, O. D., Kambeitz, J., Kim, E., Stahl, D., Slifstein, M., Abi-Dargham, A., & Kapur, S. (2012). The Nature of Dopamine Dysfunction in Schizophrenia and What This Means for Treatment: Meta-analysis of Imaging Studies. *Archives of General Psychiatry*, 69(8). <https://doi.org/10.1001/archgenpsychiatry.2012.169>
- Petersen, N., Rapkin, A. J., Okita, K., Kinney, K. R., Mizuno, T., Mandelkern, M. A., & London, E. D. (2021). Striatal dopamine D2-type receptor availability and peripheral 17 β -estradiol. *Molecular Psychiatry*, 26(6), 2038–2047. <https://doi.org/10.1038/s41380-020-01000-1>
- Zhang, J.-P., Lencz, T., & Malhotra, A. K. (2010). D 2 Receptor Genetic Variation and Clinical Response to Antipsychotic Drug Treatment: A Meta-Analysis. *American Journal of Psychiatry*, 167(7), 763–772. <https://doi.org/10.1176/appi.ajp.2009.09040598>

REVIEWER COMMENTS

Reviewer #1 (Remarks to the Author):

This is a well written paper addressing a clinically relevant and mechanistically informed, and informative issue, that is, the effects of a high dose of the antipsychotic D2 receptor antagonist sulpiride on the flexible updating of beliefs about trustworthiness. A repeated trust game with 25 trials is administered to 4 groups of subjects: one group of vulnerable A1+ carriers of the Taq1 allele, associated with altered D2 receptor density, on placebo; one A1+ group on sulpiride, one A1- group on placebo and one A1- group on sulpiride. Subjects invest money to play with a bad trustee, who defects their investment relatively often, and with a good trustee, who defects less often. The question is whether subjects learn to adapt their investments based on their experience with these trustees. Advanced mixed model analyses and hierarchical gaussian filter model-based analyses lead to the conclusion that sulpiride reduces the precision of beliefs about trustworthiness, thus increasing the weight on prediction errors. Subjects' investments vary more readily across trials after sulpiride than placebo, although they seem to converge on the same investment overall. This observation is relevant given the hypothesis that beliefs are overly rigid in psychosis, suggesting that antipsychotics might help render these beliefs more flexible.

Strengths of the paper include

- an intriguing question/timely topic that is clinically relevant,
- administration of a high dose of D2 receptor antagonist, ensuring sufficient receptor occupancy
- sophisticated Bayesian mixed model analyses of raw behaviour as well as model-based analyses
- rigorous execution of the analyses including model comparison, model validation with posterior predictive checks, simulations and parameter recovery;
- appropriate controls are in place, allowing the authors to conclude that effects on the repeated trust game do not reflect changes in e.g. the sensitivity to social feedback (as evidenced by absence of sulpiride effect on single-round social interaction tasks).

I have the following comments and questions to be addressed in a revision:

While the general rationale is clear and convincing, the basis for the directional prediction on precision weight on prediction errors in relation to the direct and indirect pathways in the introduction is unclear and hard to follow.

Answer (A): We first want to thank the reviewer for their constructive criticism of our work and for pointing out relevant gaps in our argumentation. We acknowledge the lack of clarity when formulating our central hypothesis. We have changed this part of the introduction (p.3), which now reads as follows:

“Dopamine receptors within the corticostriatal circuitry are ideally positioned to regulate the PE-related signal propagation and encode precision (Friston, 2008; Yao, Spealman, & Zhang, 2008). One possible neurobiological substrate of precision is proposed to be the post-synaptic gain of neuronal populations reporting PEs, where synaptic gain refers to the amplification or attenuation of the pre-synaptic signal on the post-synaptic cell (Adams, Stephan, Brown, Frith, & Friston, 2013; Friston et al., 2012). Post-synaptic D1 and D2 type dopamine receptors in the striatum have complementary effects on synaptic gain (Frank, 2005; Yao et al., 2008). D1-like receptors increase the excitability of post-synaptic neurons, and D2-type attenuate signal propagation and decrease synaptic gain (Frank, 2005). A prediction that comes from this is that dopamine binding to D1 receptors would promote PE propagation and increase belief updating. In contrast, D2 receptor stimulation would reduce post-synaptic responses and attenuate changes in beliefs, leading to belief rigidity (Adams, Huys, & Roiser, 2016). In line with this, when learning about others, blocking D2 receptors should increase the volatility (or rate of change) of beliefs.”

There was insufficient information in the paper about the alternative models against which the HGF model is compared, preempting evaluation of the implementation of e.g., the RW model and the associated conclusions.

This is relevant, because it was not clear why the effect does not reflect simply a higher learning rate, perhaps particularly for negative outcomes (a negative learning rate) and thus greater weight on recent versus remote (negative) outcomes. The paper would benefit from more extensive treatment of this

alternative account. While a higher learning rate might reflect less precise or less rigid beliefs about trustworthiness, it might also have various other origins.

A: We agree that a more thorough discussion of the differences between the “non-Bayesian” and “Bayesian” reinforcement learning approaches will improve the paper. We address the reviewer’s concerns by i) being more precise about the main terms we use to describe belief updating and how they relate to learning rates, ii) providing more information about the alternatives models, iii) comparing model performances more thoroughly, iv) looking at the implications of the RW model results and v) by discussing how our results relate to the RL framework in the discussion. More specifically, these are the actions we have taken:

i) There is some confusion about the term “belief volatility”, which simply describes the “rate of change” of a belief and, as such very closely linked to the concept of a learning rate (and different from “belief in the volatility of the environment” that describes participants’ beliefs that a reversal will occur). We try to resolve this by being clearer throughout the manuscript (see also our response to reviewer’s concern related to belief volatility and reversal learning below).

ii) We now extended the description of all the models in the Methods section and provided a table with all the priors used in all three models (p.27-28):

“We also compared both HGF models to a simple Rescorla-Wagner model (Rescorla & Wagner, 1972) with different learning rates for positive and negative feedback. Given a learning rate α , and an action value Q of a chosen action on trial t , we defined the updating equation as:

$$Q^{(t+1)} = Q^{(t)} + \alpha PE,$$

$$PE = BT^{(t)} - Q^{(t)}$$

while using the same likelihood function as for the HGF models. We estimated a different α for positive and negative outcomes.”

“An overview of priors of parameters across the three models can be seen in Table 3.

Table 3 | Priors for the three computational models

	HGF with γ	HGF (with κ)	RW
Hyperpriors (random effects)	κ – fixed $\mu_{\omega} \sim N(-2,1)$ $\mu_{\Delta\omega} \sim N(0,1)$ $\mu_{\eta} \sim N(10,1)$ $\mu_{\gamma} \sim N(0.05)$ $\mu_{\mu_0} \sim N(0,1)$ $\sigma \sim HN(0,0.2)$ $R \sim LKJcorr(2)$	γ – fixed $\mu_{\omega} \sim N(-2,1)$ $\mu_{\Delta\omega} \sim N(0,1)$ $\mu_{\eta} \sim N(10,1)$ $\mu_{\kappa} \sim N(0.1)$ $\mu_{\mu_0} \sim N(0,1)$ $\sigma \sim HN(0,0.2)$ $R \sim LKJcorr(2)$	$\mu_{\alpha} \sim N(0,1)$ $\mu_{\Delta\alpha} \sim N(0,1)$ $\mu_{\eta} \sim N(10,1)$ $\mu_{\mu_0} \sim N(0,1)$ $\sigma \sim HN(0,0.2)$ $R \sim LKJcorr(2)$
Priors of group-level (fixed) effects	$\beta_{*} \sim N(0,1)$ $c_k \sim N(0,5)$		

iii) We also compared the models more extensively and added a separate supplementary figure for model comparison (Supplementary Fig. 3). We describe the model comparison in the main text in the Results section (p.7):

“The RW model is a simple Q-learning model with different static learning rates for gains (positive outcomes) and losses (negative outcomes). All models used the same ordinal-logistic likelihood function and were compared based on their trial-by-trial predictive accuracy through the leave-one-out cross-validation information criterion (LOOIC) and their expected log predictive densities (ELPD). We found that the HGF model with the choice precision parameter γ outperforms both models (Supplementary Fig. 3a). We also compared the models across trials and trustees with the LOOIC and by looking at the correlations of predicted investments with actual behaviour (Supplementary Fig. 3b, c). Interestingly, the performance of both models varies

similarly across trials, with the HGF performing better across the whole task, particularly for investments against the good trustee.”

Supplementary Fig. 3 | Model comparison. **a**, Comparing the HGF model with all four parameters ($\omega, \mu_0, \eta, \gamma$) to an HGF model that includes a coupling parameter κ , but does not include γ , and to an RW model. On the y-axis is the relative difference in the predictive density (on a log scale). **b**, The HGF model with γ and the RW model compared with the leave-one-out-information criterion across trials and trustees (lower values imply a better fit). **c**, Same two models compared based on the correlations of actual investments and investments predicted by the model (higher values indicate a better fit).”

iv) We looked at the implications of the RW model. We provided another supplementary figure to display the RW model results (Supplementary Fig. 5). In short, the RW model also implies (though statistically with in part only marginal significance and lower effects than the HGF) higher learning rates, implying that even a model with a static learning rate would capture these effects. This is not surprising given that already model-free analysis detects increased changes in investments that reflected back-transfers from the trustees. We summarize the findings of the RW model in the Results section (p.8):

“We then compared the results from the HGF model to those of the RW model (Supplementary Fig. 5). We found effects in the same direction, whereby sulpiride led to (marginally) higher learning rates overall ($d = 0.315$, 95% CrI [-0.087, 0.729], $P(d < 0) = 0.064$, Supplementary Fig. 5a), an effect that the A1+ participants drove ($d = 0.593$, 95% CrI [0.03, 1.16], $P(d < 0) = 0.02$), and was not present in the A1- participants ($d = 0.037$, 95% CrI [-0.524, 0.605], $P(d < 0) = 0.453$), with a marginally significant difference between the effect ($d = -0.553$, 95% CrI [-1.343, 0.235], $P(d > 0) = 0.079$). Further, the effect of the drug in the A1+ group was observed both when learning about positive outcomes ($d = 0.697$, 95% CrI [0.029, 1.369], $P(d < 0) = 0.021$, Supplementary Fig. 5b) as well as negative outcomes ($d = 0.488$, 95% CrI [-0.059, 1.044], $P(d < 0) = 0.039$, Supplementary Fig. 5c). However, the difference across the two types of learning rates was not as pronounced as in the HGF model.

Supplementary Fig. 5 | Sulpiride effects on the RW model's learning rate. a, Average learning rate across trials. **b,** Learning rate for positive outcomes. **c,** Learning rate for negative outcomes. Below are mean effect sizes with 50% and 95% CrI."

v) We generally discuss how our results fit within the Go/No-Go RL framework and how we understand the results from the RW model in the discussion (p.12):

"When interpreting our findings within the Go/No-Go framework, it should be noted that in the repeated Trust game in this study, participants have no agency over the valence of the outcome (positive or negative back-transfer), and investments are possible only on an ordinal scale. Similarly, the RW model we used in our study should also be interpreted with this in mind. Mirroring the effects in the HGF model, sulpiride increased learning rates in the RW model for both positive and negative outcomes. In multi-arm bandit tasks or Go/No-Go tasks where the reinforcement learning framework is often used to explain choice selection, the learning rate reflects the "law of effect" whereby actions that lead to positive (negative) outcomes are more (less) likely to get repeated (Rescorla & Wagner, 1972). A positive outcome following a specific investment choice will lead to higher investment (if possible). A higher learning rate simply reflects the change in the expected response of the trustee. It is therefore related to the degree to which beliefs about trustworthiness change (on average across trials). Generally, the crucial distinction between RW and Bayesian models is that the latter assumes that agents consider the uncertainty of outcomes when updating beliefs. With this, Bayesian models such as the HGF or the Kalman filter can account for phenomena for which non-Bayesian reinforcement learning models fail, such as latent inhibition and sensory preconditioning (Gershman, 2015, 2018). Given the relative increase in the overall and trial-by-trial predictive performance of the HGF model that includes choice uncertainty over the RW, our data support the notion that uncertainty about the outcomes and what actions to take affects choice behaviour in the repeated trust game."

A (related) question is whether the pattern of (changes in) investments is better accounted for by a model that considers effects on working memory strategies. This is pertinent given sulpiride's established effects on working memory and the well-recognized finding that changes in RL in schizophrenia can reflect changes in working memory (e.g. Collins et al., 2014 and Collins et al., 2017). Might the drug have changed reliance on working memory versus reinforcement learning to elicit the

increase in learning rate (see work by Collins and Frank)? To explore this alternative explanation, the authors could consider assessing changes in investment as a function of delay, but also recognize it in the discussion if applicable.

A: Working memory plays a vital role in belief updating and the comment to try to account for working memory effects is a great idea. Unfortunately, looking at the impact of delays was not possible. Trials were pseudorandomized, whereby there were maximally two consecutive trials against the same trustee, with many participants having no trials with delays. However, one of the behavioural tasks included in the behavioural battery was also a spatial working memory task (published in Naef et al. 2017).

As reported by Naef et al. 2017, sulpiride decreased performance in both genotype groups in more challenging spatial working memory task trials. Naef et al. also note that overall memory performance was higher in the A1+ group compared to the A1- group, regardless of the drug taken. To see to what degree the working memory data affects our inference of group-level effects, we reran the hierarchical model without drug data and included only spatial working memory (WM) performance.

In line with Collins' work (cited by the reviewer) we see that the poorer model performance led to higher belief volatility estimates and choice uncertainty. However, when looking at the residuals of that model, we see that the group differences in sulpiride and placebo are still present. When we reran the model with WM, drug and genotype variables, we noticed that the inference on drug effects was not affected much.

The importance of looking at WM data is now motivated in the introduction (p.4-5), and the results of the WM analysis are displayed in a new section (p.9-10) that includes a figure (Figure 6):

“Working memory does not explain effects of sulpiride on belief updating

In the repeated trust game, participants must remember the trustees' responses to previous trials. Higher choice stochasticity could therefore be due to poorer working memory. Furthermore, the inability to remember outcomes of past trials might increase the reliance on the previous trial and thereby cause increased learning rates and belief volatility. To see to what degree our findings are influenced by the possible effects of sulpiride on working memory, we included the data from a spatial working memory (WM) task performed in the same sample and published before (Naef et al., 2017). In the spatial WM task, participants need to uncover 'tokens' from sets of boxes, whereby they need to remember which boxes were previously searched and what the outcomes of those searches were (see Methods for details). As Naef et al. report, sulpiride had a detrimental effect on working memory performance, whereby participants in both genotype groups performed more errors (opened boxes previously already opened) in more challenging task trials (trials with 10 or 12 boxes).

In this study, we first wanted to investigate whether the model parameters are influenced by WM performance. To do so, we re-estimated the hierarchical model that included only WM data at the group level without the drug and genotype variables. As can be seen from Fig. 6a, the belief volatility parameter ω was associated with a higher number of errors in the WM task ($d = 0.658$, 95% CrI [0.334, 1.01], $P(d < 0) < 10^{-3}$), and the Choice precision parameter γ negatively correlated with the number of errors ($d = -0.811$, 95% CrI [-1.087, -0.541], $P(d > 0) < 10^{-3}$). This implies that poorer working memory performance is related to higher choice and belief uncertainty. Importantly, however, when plotting the residuals of the model parameters (unexplained variance after accounting for WM effects), we see that the impact of sulpiride on belief volatility in the A1+ group is still present (Fig. 6b). To obtain posterior distributions of drug and genotype effects after accounting for WM data, we re-estimated the parameters of the model this time including WM data as well as drug and genotype variables. We find that including WM information in the hierarchical model only slightly changed the inference about the effect of sulpiride on belief updating. The main effect of sulpiride on ω was now somewhat less certain with the 95% CrI including values below 0 ($d = 0.56$, 95% CrI [-0.052, 1.211], $P(d < 0) = 0.036$), but the effect in the A1+ group was still reliably present ($b = 0.852$, 95% CrI [0.116, 1.61], $P(b < 0) = 0.014$, $d = 0.694$, 95% CrI [0.093,

1.369]). Similarly, posterior intervals of sulpiride effects on γ after including WM data were comparable to those without WM data, with the main effect remaining convincingly negative ($d = -1.034$, 95% CrI [-1.634, -0.461], $P(d>0) = 0$), and the effect is more pronounced in the A1- group ($d = -1.562$, 95% CrI [-2.423, -0.722], $P(d>0) = 0$) and marginally significant in the A1+ group ($d = -0.512$, 95% CrI [-1.168, 0.133], $P(d>0) = 0.056$).

Fig. 6 | Working memory performance and computational parameters. a, We reran the parameter estimation with a multilevel model that only included working memory data (number of errors) at the group level effect (and was agnostic about drug and genotype groups). Poorer performance in the spatial working memory task correlated positively with belief volatility ω and negatively with choice precision γ and did not affect noise or initial trustworthiness. Effect sizes depicted with means, 50% and 95% CrIs. **b,** From the model that is agnostic about drug and genotype data, we plot residual variances that remain after accounting for working memory data for parameters ω and γ . In the second step, the parameters were estimated with working memory data and drug and genotype variables at the group level. The results of this analysis are shown below as effect sizes with means, 50% and 95% CrIs. The analysis is compared with the model that does not include working memory data.”

The results of this analysis are also briefly discussed on p.13:

“One important factor that could confound increased belief volatility and learning rates following sulpiride administration is working memory. Previous work has shown that individual differences in working memory capacity contribute to behavioural variability in reinforcement learning tasks (Collins, Brown, Gold, Waltz, & Frank, 2014; Collins & Frank, 2012) whereby decreased memory capacity might lead to a higher salience of more recent outcomes and therefore higher learning rates (Collins, Ciullo, Frank, & Badre, 2017). We also find support for this notion in our data, whereby poorer WM performance was strongly linked to higher belief volatility and higher choice uncertainty. However, despite sulpiride decreasing WM capacity in our cohort, including WM data in the model did not affect inference about sulpiride effect on belief

volatility, nor on choice stochasticity. This increase in choice uncertainty or stochasticity under sulpiride that is therefore likely not due to failures in working memory capacity. Instead it could have been due to participants being less motivated to maximise outcome, and therefore less likely to behave as a rational “homo economicus” (Camerer, 2003). Were that the case in our study, one would expect a different behavioural pattern in the single shot trust games. Participants under sulpiride should behave less as rational agents and therefore would be less likely to punish betrayals and reward trusting behaviour (Fehr, Fischbacher, & Gächter, 2002).”

In this same context I wondered what is the optimal learning rate in this task, and was puzzled about the data presented in supplementary figure 2, showing that subjects - on placebo - did not seem to learn to increase investments for the good trustee, while reducing investments across trials for the bad trustee. Isn't this a pattern one would want to see if subjects do the task as instructed?

While it is stated in the results section that average investments were not affected by drug or genotype, I did not see the statistics supporting this statement, and it was unclear whether this analysis was performed as a function of trial. I also suggest that beliefs for bad and good trustees are plotted as a function of trial, drug, genotype and trustee in the main text (much like Supp Fig 2 which I would think belongs to the main text too).

A: We agree that it is helpful for the readers to see the average investments in the main text. In the revised manuscript, we now include a figure (Figure 2) that displays average investments across trials (a simplified Supplementary Figure 2, which seemed too big for the main text) and average points earned. We added this section in the Results (p.6):

“D2/D3 receptor antagonism has no effect on average investment behaviour or overall performance

Next, we looked at whether this higher change of investments from one trial to the next is reflected in average investment patterns (Fig. 2a, for more detailed plots see Supplementary Fig. 2). An ordinal logistic model predicting investments from Treatment, Genotype, Trustee and Trial variables, with a random slope for Trustee and Trial showed no difference between sulpiride and placebo on average investment behaviour either in the A1+ group ($b_{good} = -0.011$, 95% CrI [-1.95, 1.796], $P(b_{good}>0) = 0.494$, $b_{bad} = 0.089$, 95% CrI [-2.256, 2.47], $P(b_{bad}<0) = 0.47$), nor in the A1- group ($b_{good} = -0.496$, 95% CrI [-2.353, 1.367], $P(b_{good}>0) = 0.297$, $b_{bad} = 0.821$, 95% CrI [-1.508, 3.257], $P(b_{bad}<0) = 0.244$). There were no differences in initial investments across the four drug and genotype groups (Supplementary Fig. 2). The overall initial investment was estimated to be, on average, 6.33 (95% CrI [3.129, 9.687]), suggesting that most participants expected a positive back-transfer initially. In line with this, the slope when playing against the good trustee was only marginally positive ($b = 0.086$, 95% CrI [-0.008, 0.187], $P(b<0) = 0.036$), while the slope when playing against the bad trustee was convincingly negative ($b = -0.153$, 95% CrI [-0.284, -0.027], $P(b>0) = 0.009$). While there were no differences between slopes across the drug groups in the A1+ participants ($b_{good} = -0.058$, 95% CrI [-0.184, 0.065], $P(b_{good}>0) = 0.179$, $b_{bad} = 0.044$, 95% CrI [-0.125, 0.21], $P(b_{bad}<0) = 0.297$), there was a slight increase in the slope following sulpiride administration in the A1- group when playing against the bad trustee ($b_{bad} = 0.159$, 95% CrI [-0.007, 0.336], $P(b_{bad}<0) = 0.03$) but not when playing against the good trustee ($b_{good} = 0.069$, 95% CrI [-0.06, 0.19], $P(b_{good}<0) = 0.14$). Similarly, there were no differences across drug and genotype groups regarding how many points they earned when playing against both trustees (Fig. 2b, Supplementary Table 13).

Fig. 2 | Effects of sulpiride on average investments in the Repeated Trust Game. a, Average investment behaviour grouped for each trustee. Lines depict model predictions (means with 95% CrI as the shaded area), plotted over raw means for each drug group (Δ). We found no main effect of sulpiride in either genotype group on average investment behaviour regardless of the trustee. For statistics, refer to the main text and Supplementary Fig. 2. **b,** Overall points earned in the task grouped for each trustee. Boxplots over average points earned for each participant.“

As can be seen from Figure 2, there were no apparent differences in average investments across groups. Notably, participants invested more than half initially, which led to less steep learning slopes for the good trustee, but all groups did learn on average. Regarding what constitutes optimal behaviour in this task, we already had a note in the supplementary (Supplementary Note 2, p.6 of Supplementary Material) where we discussed optimality from different angles. This has now been extended and reads:

“Supplementary Note 2: Optimal behaviour in the repeated trust game

Optimal behaviour in the repeated trust game can be defined within various frameworks. Within the utility maximization framework, a “rational” outcome-maximizing agent would, at each trial, choose the investment that brings the highest expected value. When the trustee betrays the investor after an investment I , the investor ends up with an outcome of $10 - I$. When the trustee equalizes, both players receive half of the overall gain. The outcome V in that case is:

$$V = \frac{I - 10 + 10 + 3 * I}{2} = 10 + I$$

If an agent has a subjective belief that the trustee will equalize with probability p , then the expected value of his investment I is:

$$EV(I) = p * (10 + I) + (1 - p) * (10 - I) = 10 + I(2p - 1),$$

which is a linear function of p . Meaning that as soon as p is believed to be above 0.5, the outcome maximizing agent should give ten and as soon as p is below 0.5 they should invest 0.

We note that in our model, an outcome-maximizing agent would have extremely high values of γ , which would also correspond to highly exploitative behaviour defined within the reinforcement learning framework.

Importantly, this outcome-maximizing agent does not consider the uncertainty about their estimate. On the other hand, a Bayesian perspective defines behaviour in this task as optimal given the investor’s (participant’s) subjective prior beliefs (Daunizeau et al., 2010). Update equations of the model are determined deterministically from the free parameters and describe Bayes optimal belief trajectories (Mathys, Daunizeau, Friston,

& Stephan, 2011). For example, if one's prior is that other people's behaviour is highly volatile, then it is optimal to adjust your belief after each feedback, whereas if one expects individuals to behave consistently, one should form an opinion quickly. With this, the Bayesian framework can describe subjectively optimal behaviours but objectively maladaptive (Mathys et al., 2011)."

Critically, I would have thought that a hypothesis specifically about flexible belief updating is better addressed using a paradigm in which trustworthiness fluctuates across time at different frequencies (e.g. with some volatile task phases where the good trustee becomes a bad one and vice versa relatively frequently, and some stable task phases where contingencies are more stable). It is that type of setting that would allow a cleaner isolation of effects on learning rate per se and an effect on belief volatility.

A: This is an important point, and we thank the reviewer for encouraging us to clarify this! We acknowledge that *Belief flexibility* and *Belief volatility* might both be a bit ambiguous terms. Belief volatility in our manuscript measures how much a belief changes over time as new information comes in. This is different from beliefs about how volatile the environment is (that could be described as "belief in the volatility of the environment or task contingencies") and is usually assessed in learning tasks with reversals. Although usually, HGF models also estimate "environmental volatility" and how beliefs about it change through time, our model assumes that the environment is stable. Admittedly, belief flexibility is usually used when subjects need to adapt their beliefs to latent changes (reversals). To avoid confusion, we now try to be more precise throughout the manuscript. Some of these changes include the following:

We changed the first sentence of the abstract:

~~"The ability to flexibly adjust beliefs about other people is crucial for human social functioning.~~

The ability to learn about other people is crucial for human social functioning."

In the introduction (p.2), we now define what we mean by belief volatility:

"As in similar computational frameworks (Sutton & Barto, 2017), the belief update is proportional to the deviation of the prediction from the actual outcome, termed a prediction error (PE), weighted by the precision of prior beliefs. On top of this, new information also reduces uncertainty about the outcome. When prior beliefs are highly uncertain, the weight on the PE will be high, and beliefs will be highly volatile. Conversely, if beliefs are held with high precision, this leads to a down-regulation of the influence of PE on learning and lowers belief volatility. Inflexibility in forming beliefs about others proportionally to their actions can result from high precision of prior beliefs about others' attitudes. Yet, the neurocomputational and neurochemical mechanisms regulating the uncertainty of beliefs are poorly understood. In this study, we examined the effects of the antipsychotic drug sulpiride, a D2/D3 dopamine receptor antagonist, on the uncertainty of beliefs about another person's trustworthiness."

Finally, at the end of the introduction (p.4), we explicitly mention that our design did not focus on task volatility:

"Importantly, we told the participants that the other players had given their answers weeks before the study day; therefore, their decision to equalize or betray did not depend on the participant's investment. With this procedure, we increased the likelihood that their investments reflected the degree of uncertainty they had about the other player's response and were not confounded by strategic investment strategies or exploratory action policies. By asking the participants to learn about a stable feature, we also ensure that participants' behaviour did not reflect differences in beliefs in the task volatility (the likelihood that the other person changed their mind), which might have obscured more basic processes related to forming beliefs about others."

Some example trial sequences are plotted in Figure 2b, but it is rather opaque what parameters are plotted on the y-axis. I suggest extending the legend considerably. And plotting the across-subject-average timeseries of investments for bad and good trustees (as suggested above, and in Supp Fig 2).

A: We agree that the figure was not clear and thank the reviewer for this suggestion! We have extended this figure (now Figure 3) and added Figure 2 with average investment plots.

Finally, the observation at the end of the abstract and discussion section that the findings support the view that antipsychotics help to reduce the impact of distressing rigid beliefs seems to represent a jump-to-conclusions, particularly because the task does not involve reversals, or address the flexible updating of beliefs that are (now) inadequate. I note also that the paragraph preceding this last point on impairments in the formation of internal models, yet enhanced reliance on those internal models is a bit confusing.

A: We generally agree that the translation of these findings to patient treatment is not straightforward. We have changed this paragraph of the discussion (p.13-14) and adapted the concluding sentence of the abstract accordingly.

“Our findings might be particularly relevant for understanding the effects of antipsychotic medication in patients with psychosis, a disorder characterized by rigid beliefs of persecution, underlined by a profound lack of trust in others (Freeman, 2016; Fuchs, 2015). Previous studies with repeated trust games showed that patients with psychosis have lower initial trust and find it hard to change their beliefs (Fett et al., 2012; Gromann et al., 2013). Neurocomputational accounts of delusions suggest that hyperactivity of D2 receptors in patients leads to increased precision beliefs that result in rigid convictions held with high confidence (Adams et al., 2013; Sterzer et al., 2018) and a recent paper shows that higher belief instability in patients with schizophrenia predicts responses to psychotherapeutic treatment (Hauke et al., 2022). This suggests that decreasing belief rigidity through D2 antagonism could be an essential contributor to the success of adjunct psychosocial treatment. However, there are profound differences between the effects of repeated use of antipsychotics in patients and acute D2 antagonism in healthy volunteers. For example, rodent studies show that although in healthy animals D2 antagonisms increases the activity of midbrain dopaminergic cells, this pattern is reversed in an established animal model of schizophrenia (Valenti, Cifelli, Gill, & Grace, 2011).

Furthermore, despite rapid receptor blockade of D2 antagonists, the inhibition of excessive dopaminergic signalling proposed to underly the therapeutic effects comes only after weeks of treatment (Grace, Bunney, Moore, & Todd, 1997; Kapur, Agid, Mizrahi, & Li, 2006). Our data also suggest that the therapeutic effect could be larger in patients that are A1 allele carriers of the Taq1a polymorphism. Yet there is no evidence for this (Zhang, Lencz, & Malhotra, 2010), despite higher dopamine synthesis being the most likely biomarker of psychotic symptoms (Fusar-Poli & Meyer-Lindenberg, 2013; Howes et al., 2012) and a predictor of response to antipsychotic treatment (Jauhar et al., 2019). Translating our findings to clinical practice will require more work with targeted patient populations.”

Other comments:

Please specify in the paper whether the trial sequence is fixed pseudorandom, that is the same for both sessions and for all participants?

A: Good point. We do that now when describing the task in the Methods section (p.23):

“The feedback was pseudorandomized separately for each participant and was interleaved whereby only two consecutive trials with the same trustee were allowed.”

What was the delay between drug administration and administration of this specific task? What is time to peak for 800mg sulpiride.

A: Peak plasma sulpiride levels occur after three hours, which is the length of our waiting period. The behavioural testing began around 200 minutes post administration and started with the repeated trust game. This is now also mentioned in the manuscript in the Methods/Procedure section (p.22).

Would one not have expected subjective effects of the high dose of sulpiride, e.g. on alertness and mood?

A: We report side effects in the supplementary material and note that there were no marked differences between groups in blood-pressure, heart-rate, self-reported side effects and mood, including alertness. The only significant effect ($p = 0.049$) was that after sulpiride participants were more talkative, which we do not deem relevant to our conclusions.

Sample sizes are somewhat modest, a point that might be emphasized particularly given the lack of an available preregistration

A: We agree and mention this as a limitation in the discussion (p.14):

“Several important limitations should be kept in mind when considering the generalizability of the findings in this study. First, the sample was limited to male participants. This restriction was initially motivated by the notion that including female participants would require more than doubling the sample size due to increased variance of dopamine availability across the ovarian cycle. However, recent work shows no support for this (Petersen et al., 2021). Given that there are important sex differences in responses to antipsychotics, both in terms of efficacy and side-effect profiles (Hoekstra et al., 2021), future work should prioritize studies in females. Second, despite our hypothesis-driven approach, the sample size of the genetic subgroups was small. The drug-gene interactions we report should be interpreted with this in mind. We note however that we do find a main effect of sulpiride on increased investment change and on belief volatility in a sample size that is comparable to other pharmacological studies with a between-subject design (Martins, Mehta, & Prata, 2017).”

Reviewer #2 (Remarks to the Author):

In this study, the authors investigate how the administration of a selective D2/D3 dopamine receptor antagonist (sulpiride) impacts how we learn about the trustworthiness of others. Using a repeated trust game and a Bayesian belief model, the authors found that the administration of sulpiride led to (1) a higher uncertainty of the beliefs about the trustworthiness of others, but only in participants with at least one minor A1 allele of the Taq1a polymorphism (associated with higher endogenous dopamine levels); and (2) a higher uncertainty about investment selection, in both groups. Overall, the paper is very well written and meets the standards for publication. This work represents an important and relevant research topic in the field, the results are convincing and clearly presented. I find no errors per se, and I only have a few minor remarks and clarification suggestions for the authors.

Answer (A): We thank the reviewer for their positive feedback and for providing us with several relevant points that improve the clarity and comprehensibility of the manuscript.

1) Page 2 § 2. In the introduction, the authors state “For instance, if we firmly believe someone is hostile towards us, a positive gesture coming from them will not much change our belief about them. On the other hand, that same gesture from someone whose intentions we are unsure of, will likely strongly shift what we think about them.” I suggest that this should be phrased as an assumption since it is not supported by any reference and could even go against the findings from Siegel et al 2018, which were not cited by the authors but seem to be relevant for their study (in addition, the authors seem to find a similar pattern as described on page 6 § 4).

A: This sentence refers to the reference in the previous sentence alluding to the notion that when we are uncertain about others’ attitudes, their actions matter more (regardless of whether it is a positive or negative gesture). We have adapted the sentence to clarify this point (p.2) hopefully:

“How behaviours of others affect our overall beliefs about their trustworthiness largely depends on how certain we are about the attitudes that presumably drive others’ actions (FeldmanHall & Shenhav, 2019). For instance, if we believe someone will be hostile or friendly towards us with high certainty, any gesture from them will not much change our belief about them. On the other hand, that same gesture from someone whose intentions we are unsure of will likely strongly shift what we think about them.”

That being said, we agree that the asymmetry of belief uncertainty regarding good and bad intentions, as described in Siegel et al., is interesting and relevant. We now also report the main effect of the trustee on belief volatility in the results and discuss this in the discussion.

The corresponding section in the results (p. 8) now reads:

“Looking at potential asymmetries when dealing with uncertainty around beliefs about trustworthy or untrustworthy partners, we find that on average, the volatility of beliefs about the bad trustee were more volatile ($d = 0.412$, 95% CrI [0.031, 0.827], $P(d < 0) = 0.018$). When examining the drug effects, we observed that in the A1+ group, the difference in ω between placebo in sulpiride is apparent in interactions with both trustees ($d_{\text{bad}} = 1.62$, 95% CrI [0.731, 2.644], $P(d_{\text{bad}} < 0) < 0.001$, $d_{\text{good}} = 0.689$, 95% CrI [-0.089, 1.575], $P(d_{\text{good}} < 0) = 0.042$, but is higher for the bad trustee ($d_{\text{good-bad}} = -0.923$, 95% CrI [-1.754, -0.121], $P(d_{\text{good-bad}} > 0) = 0.01$, Supplementary Fig. 4b, c). Interestingly, this analysis also showed that in the A1- group, there is a significant interaction of sulpiride and trustee effects ($d_{\text{good-bad}} = -1.453$, 95% CrI [-2.529, -0.51], $P(d_{\text{good-bad}} > 0) = 0.001$, Supplementary Fig. 4b, c), whereby in that genetic group the effects of sulpiride on belief volatility are marginally significant for the bad trustee ($d_{\text{bad}} = b = 0.711$, 95% CrI [-0.21, 1.669], $P(d_{\text{bad}} < 0) = 0.066$), but even negative for the good trustee ($d_{\text{good}} = -0.742$, 95% CrI [-1.74, 0.091], $P(d_{\text{good}} > 0) = 0.042$). At this point, we also note that on average, participants expected the trustee to reciprocate ($d = 0.717$, 95% CrI [0.406, 1.023], $P(d < 0) = 0$) with initial inferred probability of reciprocation being 0.67 (95% CrI [0.60, 0.73]). However, there were no differences between treatment groups in initial beliefs (μ_0) about the trustworthiness either overall ($d = -0.166$, 95% CrI [-0.697, 0.37], $P(b > 0) = 0.272$, Supplementary Fig. 4), in the A1+ group ($d = 0.111$, 95% CrI [-0.585, 0.802], $P(d < 0) = 0.383$), or in the A1- group ($d = -0.441$, 95% CrI [-1.164, 0.287], $P(d > 0) = 0.112$).”

This result is discussed on p.12:

“One fundamental distinction that separates risky decision-making under asocial compared to social conditions is an aversion to betrayal (Fehr, 2009). People are less risk-taking in social interactions and might be particularly sensitive to indications of untrustworthy interaction partners (Bohnet, Greig, Herrmann, & Zeckhauser, 2008). Using a similar model to ours, previous work has shown that belief volatility was higher when assessing (morally) bad agents (Siegel, Mathys, Rutledge, & Crockett, 2018). Our study also found higher belief volatility when playing against the bad trustee. Although the drug effects on belief volatility were present across both trustees, the effects were stronger for the bad trustee. One reason for this could be that because participants initially expected higher trustworthiness and higher rates of positive back transfers, there was more to learn when playing against the bad trustee and, therefore, more variance across investment behaviour. This asymmetric increase in sensitivity to negative outcomes would also be in line with the notion that D1 and D2 receptors in the striatum contribute to positive and negative outcome processing via the “Go” and “No-

Go” pathways, respectively (Frank & O’Reilly, 2006; Frank, Seeberger, & O’Reilly, 2004). According to this circuit model, D2 antagonism mimics the dopaminergic dip that occurs in negative reinforcement and therefore enhances learning to avoid action with a negative outcome. However, contrary to what we observed, this model also predicts that blocking post-synaptic D2 receptors should decrease positive prediction error propagation.”

2) Page 5 § 3. The definition of parameters appears like a bullet list and leaves the reader to wonder why each of them is truly relevant in the model. It is specified later that each parameter describes a specific behavioral pattern, and this was supported by the model comparison, but it is not clear from this part of the manuscript.

A: We acknowledge that this paragraph did not sufficiently motivate the choice of parameters. However, the primary role of this paragraph was to bridge the behavioural and computational modelling analysis; we felt this was not the place for a more detailed parameter description. We have therefore shortened this paragraph and provide a more thorough explanation of the parameters in the first paragraph of the Results/Computational framework section (p.6-7):

“The belief model uses a hierarchical Gaussian filter (HGF) to generate trial-wise sequences of participants’ beliefs about the trustworthiness of two trustees as well as the uncertainty (or precision) surrounding those beliefs (Fig 3, (Mathys et al., 2011, 2014)). We estimated a participant-specific parameter ω , called *belief volatility*, that describes how each participant’s precision of beliefs evolved over time and consequently determined the relative rigidity (or malleability). More specifically, on each trial, we approximate the latent belief about the trustworthiness of the other player as a gaussian distribution with a specific mean and variance. Higher belief volatility ω implies higher variance (or lower precision) of trial-by-trial belief estimates. Importantly, the dynamic learning rate (ψ_t) on the PE is proportional to the expected variance or inversely proportional to the precision of beliefs and is therefore referred to as a “precision-weight”. Low precision of prior beliefs leads to higher precision-weighted learning rates and stronger shifts in beliefs throughout the task (see two example belief trajectories with different ω values in Fig. 3b).

The beliefs about trustworthiness are mapped on to probability of positive or negative feedback with an inverse logistic function. Because D2 receptor activity is linked to choice uncertainty and action variability (Adams et al., 2020; Eisenegger et al., 2014), we also included another parameter called choice precision parameter γ that determined the nonlinear mapping from beliefs to the investments. Higher choice precision implies an investment distribution centred around extremes (i.e., investing 0 and 10), and lower values imply a more dispersed investment distribution and more uncertainty or stochasticity in action selection. It thus mirrors the stochastic aspect of the inverse temperature parameter in the softmax equation often used in non-ordinal (e.g., binary) choice tasks. Finally, how beliefs about the probability of a positive back-transfer affect investment behaviour is determined through an ordinal logistic likelihood function. The degree to which inferred trustworthiness correlates with investments is determined by the final parameter called the trustworthiness slope (η). Crucially, the computational parameters of the model represent distinct behavioural patterns and can be recovered reliably (Fig. 3c). To determine how noisy trials are represented in the model, we defined mistake trials as trials where participants either decreased their investment after a positive back-transfer or increased their investment after a betrayal (for exact definition see Supplementary Note 1). Importantly, we observed that belief volatility ω correlates with reciprocity ($r = 0.277$, $t = 2.476$, $df = 74$, $p = 0.016$, Fig. 3d) and that the log-transformed choice uncertainty parameter γ

correlates negatively with the proportion of mistake trials ($r = -0.592$, $t = -6.3254$, $df = 74$, $p < 10e-3$, Fig. 3d) implying higher randomness in investment selection. We also predicted data from the posterior distributions of parameters. We confirmed that the model captures the crucial aspects of behaviour (Fig. 3e,f) and plotted average beliefs about the other player's trustworthiness, grouped for each trustee. (Fig. 3g)."

3) Page 6 § 1. The acronym "HGF" needs to be defined here as well, in addition to the legend of Figure 2 and the Methods section.

A: Thank you for pointing this out. We now define the acronym in the first paragraph of the Results/Computational Framework section (p. 6).

4) Page 6 § 3. This is my main concern about the readability of the paper: I don't think the precision weights have been properly defined so far. Does it refer to the weight on PE or on prior beliefs? In which case, how does this relate to learning rates? The precision weights are such an important part of the results that they should be more clearly defined, as well as the consequences of them being higher or lower.

A: We agree that there should be no ambiguity when it comes to the definition of precision-weights. Apart from being more precise (see the paragraph in answer to reviewer's point 2), we also adapted the figure that introduces the computational framework (now Figure 3):

Fig. 3 | Computational modelling. **a**, We defined a generative model that describes the evolution of participants' beliefs about the other person's trustworthiness as a Gaussian random walk with the step size of ω . The hierarchical Gaussian filter (HGF) inverts this model and provides trial-level estimations of participants' beliefs about the trustworthiness of others as Gaussian variables with mean $\mu^{(t)}$ and standard deviation $\sigma^{(t)}$. The evolution of the variances $\sigma^{(t)}$ is determined by the belief volatility parameter ω . The precision-weights $\psi^{(t)}$ are proportional to $\sigma^{(t)}$ and serve as dynamic learning rates when updating beliefs about the trustworthiness of the other player. We also estimate initial trustworthiness belief per participant (μ_0). The ordinal logistic link

function governs how beliefs about others' trustworthiness map to investments with two additional subject-level parameters: choice uncertainty (γ) and the slope (η). The parameter estimation is done through hierarchical Bayesian inference, where we estimate all individual and group-level parameters in one inferential step. **b**, Two example belief trajectories portray the different behaviours that the model can capture. The participants have different belief volatilities for the good (ω_{good}) and the bad trustee (ω_{bad}). Higher ω implies more uncertainty surrounding the trustworthiness beliefs ($\sigma^{(t)}$), which in turn leads to stronger belief shifts. **c**, For each participant, we randomly draw parameters from their individual posterior distribution, simulate data, and re-estimate them five times. Relative high correlations indicate that the model parameters are well-defined. **d**, The two main parameters of interest, belief volatility and choice uncertainty, correlate with distinct behavioural features. **e-f**, Posterior predictive for **(e)** absolute investment change from one trial to the next and **(f)** for the average investment behaviour. Plotted over raw means per trial per group and with standard deviations of predictions in the shaded area. **g**, Average beliefs about the trustworthiness (μ), with average uncertainty around the investment (σ)."

5) Page 8 § 3. Some parts of the discussion will benefit from a more cautious formulation or interpretation. For example, we know from several studies (e.g., FeldmanHall et al 2015 involving matched gambling and trust game tasks) that participants exhibit different behavioral patterns, including uncertainty valuation, in social vs. non-social tasks. Therefore, the results found in this social trust game might not be applicable to non-social reinforcement learning tasks.

A: We agree and have added a paragraph in the discussion addressing his point (p.12):

"One fundamental distinction that separates risky decision-making under asocial compared to social conditions is aversion to betrayal (Fehr, 2009). People are less risk-taking in social interactions and might be particularly sensitive to indications of untrustworthy interaction partners (Bohnet et al., 2008). Using a similar model to ours, previous work has shown that belief volatility was higher when assessing (morally) bad agents (Siegel et al., 2018). In our study we also find higher belief volatility when playing against the bad trustee. Although the drug effects on belief volatility were present across both trustees, the effects were stronger for the bad trustee. One reason for this could be that because participants initially expected higher trustworthiness and higher rates of positive back transfers, there was more to learn when playing against the bad trustee and therefore more variance across investment behaviour. This asymmetric increase in sensitivity to negative outcomes would also be in line with the notion that D1 and D2 receptors in the striatum contribute to positive and negative outcome processing, via the "Go" and "No-Go" pathways, respectively (Frank & O'Reilly, 2006; Frank et al., 2004). According to this circuit model, D2 antagonism mimics the dopaminergic dip that occurs in negative reinforcement and therefore enhanced learning to avoid an action with a negative outcome. However, contrary to what we observed, this model also predicts that blocking post-synaptic D2 receptors should decrease positive prediction error propagation."

6) Page 9 § 2 and related to point 5. The statement "suggesting that the investment selection under sulpiride was not random, but only less precise" is quite difficult to grasp and could be reformulated. It is unclear compared to the first paragraph of the discussion, where a variation of action selection parameter led to more or less stochastic choices.

A: The ending sentence of the paragraph in the discussion (p.12) now reads:

”It is possible that the increased action variability resulted from reduced motivation or increased noise in belief updating and not in choice selection per se (Findling, Skvortsova, Dromnelle, Palminteri, & Wyart, 2019). What speaks against this interpretation is that the overall performance in the task was not reduced following sulpiride administration for either of the genetic subgroups, suggesting that the investment selection under sulpiride was not random and instead reflected uncertainty about which investment to choose when interacting with the other player.”

7) I suggest the authors define the variables in Supplementary Note 1.

A: We have clarified the variables in Supplementary Note 1 as suggested:

“Supplementary Note 1: Definition of reciprocal and mistake trials

To avoid ambiguity, we note the exact definition of Reciprocal and Mistake Trials used in the main text. Denoting *Backtransfer* as a variable for positive (1) or negative (-1) feedback from the trustee, and *Change* as a variable for the relative change in investment from the previous trial, we defined Reciprocal and Mistake Trials as follows:

$$\begin{aligned} \text{Reciprocal Trial} &= ((\text{Backtransfer} == 1 \ \& \ \text{Change} > 0) | (\text{Backtransfer} == 1 \ \& \ \text{Investment} == 10)) | ((\text{Backtransfer} == -1 \ \& \ \text{Change} < 0) | (\text{Backtransfer} == -1 \ \& \ \text{Investment} == 0)), \\ \text{Mistake Trial} &= ((\text{Backtransfer} == 1 \ \& \ \text{Change} < 0) | (\text{Backtransfer} == -1 \ \& \ \text{Change} > 0)) \end{aligned}$$

References

- FeldmanHall O, Raio CM, Kubota JT, Seiler MG, Phelps EA. The Effects of Social Context and Acute Stress on Decision Making Under Uncertainty. *Psychol Sci.* 2015 Dec;26(12):1918-26. doi: 10.1177/0956797615605807. Epub 2015 Nov 5. PMID: 26546080; PMCID: PMC4679610.
- Siegel JZ, Mathys C, Rutledge RB, Crockett MJ. Beliefs about bad people are volatile. *Nat Hum Behav.* 2018 Oct;2(10):750-756. doi: 10.1038/s41562-018-0425-1. Epub 2018 Sep 17. PMID: 31406285.

Reviewer #3 (Remarks to the Author):

Mikus and colleagues present a study that uses pharmacological antagonism of dopamine D2-like receptors to evaluate the role of dopamine in trusting behavior with a specific interest in belief updating. They apply a hierarchical gaussian filter to model choices made in the trust game and show that the D2-like receptor antagonist sulpiride increased choice uncertainty, the volatility of beliefs, and the extent to which this volatility influenced investments. The authors identified specific associations that depend on Taq1A genotype polymorphism, indicating that endogenous DA is an important moderator of general precision coding. I applaud the authors' for their careful consideration of alternative explanations and important checks in the form of modified versions of the task and comparison of their model to a simple learning model. The relation between dopamine and decisions in social contexts are greatly understudied, so this work is a welcomed contribution to the study of the neurobiology of decision making. I found the description of the modeling and methods to be clear. In general, I have few major concerns.

Major Comments:

1.) I am concerned about unintended consequences of a high dose of sulpiride. The Mehta et al., 2008 work cited, for instance, indicated that even 400 mg produces working memory deficits. Are these the same subjects reported in Naef et. al., 2017 also showing working memory impairment from 800 mg of sulpiride? Can the authors provide any assurance that such deficits could not account for the observed increased belief volatility or increase in choice uncertainty? Work by Anne Collins & Michael Frank, for instance, has shown that working memory is a contributor to reinforcement learning. Would such explanations affect interpretation of the current findings if the observed effects of sulpiride are working-memory related? In general, I am concerned whether working memory capacity may be more central to the mechanisms here than is currently discussed.

Answer (A): First, we would like to thank the reviewer for their positive feedback on our study and the constructive comments to improve it.

We agree that WM plays an essential role in learning and the repeated trust game in and the point that WM may influence our findings is well taken, especially in light of findings from Naef et. al., 2017. We have now included the WM data from Naef et al. 2017 in the manuscript. As reported by Naef et al. 2017, sulpiride decreased performance in both genotype groups in more challenging spatial working

memory task trials. Naef et al. also note that overall memory performance was higher in the A1+ group compared to the A1- group, regardless of the drug taken. To see to what degree the working memory data affects our inference of group-level effects, we reran the hierarchical model without drug data and included only spatial working memory (WM) performance.

In line with Collins' work (cited by the reviewer) we see that the poorer model performance led to higher belief volatility estimates and choice uncertainty. However, when looking at the residuals of that model, we see that the group differences in sulpiride and placebo are still present. When we reran the model with WM, drug and genotype variables, we noticed that the inference on drug effects was not affected much.

The importance of looking at WM data is now motivated in the introduction (p.4-5), and the results of the WM analysis are displayed in a new section (p.9-10) that includes a figure (Figure 6):

“Working memory does not explain effects of sulpiride on belief updating

In the repeated trust game, participants must remember the trustees' responses to previous trials. Higher choice stochasticity could therefore be due to poorer working memory. Furthermore, the inability to remember outcomes of past trials might increase the reliance on the previous trial and thereby cause increased learning rates and belief volatility. To see to what degree our findings are influenced by the possible effects of sulpiride on working memory, we included the data from a spatial working memory (WM) task performed in the same sample and published before (Naef et al., 2017). In the spatial WM task, participants need to uncover 'tokens' from sets of boxes, whereby they need to remember which boxes were previously searched and what the outcomes of those searches were (see Methods for details). As Naef et al. report, sulpiride had a detrimental effect on working memory performance, whereby participants in both genotype groups performed more errors (opened boxes previously already opened) in more challenging task trials (trials with 10 or 12 boxes).

In this study, we first wanted to investigate whether the model parameters are influenced by WM performance. To do so, we re-estimated the hierarchical model that included only WM data at the group level without the drug and genotype variables. As can be seen from Fig. 6a, the belief volatility parameter ω was associated with a higher number of errors in the WM task ($d = 0.658$, 95% CrI [0.334, 1.01], $P(d < 0) < 10^{-3}$), and the Choice precision parameter γ negatively correlated with the number of errors ($d = -0.811$, 95% CrI [-1.087, -0.541], $P(d > 0) < 10^{-3}$). This implies that poorer working memory performance is related to higher choice and belief uncertainty. Importantly, however, when plotting the residuals of the model parameters (unexplained variance after accounting for WM effects), we see that the impact of sulpiride on belief volatility in the A1+ group is still present (Fig. 6b). To obtain posterior distributions of drug and genotype effects after accounting for WM data, we re-estimated the parameters of the model this time including WM data as well as drug and genotype variables. We find that including WM information in the hierarchical model only slightly changed the inference about the effect of sulpiride on belief updating. The main effect of sulpiride on ω was now somewhat less certain with the 95% CrI including values below 0 ($d = 0.56$, 95% CrI [-0.052, 1.211], $P(d < 0) = 0.036$), but the effect in the A1+ group was still reliably present ($b = 0.852$, 95% CrI [0.116, 1.61], $P(b < 0) = 0.014$, $d = 0.694$, 95% CrI [0.093, 1.369]). Similarly, posterior intervals of sulpiride effects on γ after including WM data were comparable to those without WM data, with the main effect remaining convincingly negative ($d = -1.034$, 95% CrI [-1.634, -0.461], $P(d > 0) = 0$), and the effect is more pronounced in the A1- group ($d = -1.562$, 95% CrI [-2.423, -0.722], $P(d > 0) = 0$) and marginally significant in the A1+ group ($d = -0.512$, 95% CrI [-1.168, 0.133], $P(d > 0) = 0.056$).

Fig. 6 | Working memory performance and computational parameters. a, We reran the parameter estimation with a multilevel model that only included working memory data (number of errors) at the group level effect (and was agnostic about drug and genotype groups). Poorer performance in the spatial working memory task correlated positively with belief volatility ω and negatively with choice precision γ and had no effect on noise or initial trustworthiness. Effect sizes depicted with means, 50% and 95% CrIs. **b**, From the model that is agnostic about drug and genotype data we plot residual variances that remain after accounting for working memory data for parameters ω and γ . In the second step, the parameters were estimated with both working memory data as well as drug and genotype variables at the group level. Results of this analysis is shown below as effect sizes with means, 50% and 95% CrIs. The analysis is compared with the model that does not include working memory data.”

The results of this analysis are also briefly discussed on p.13:

“One important factor that could confound increased belief volatility and learning rates following sulpiride administration is working memory. Previous work has shown that individual differences in working memory capacity contribute to behavioural variability in reinforcement learning tasks (Collins, Brown, Gold, Waltz, & Frank, 2014; Collins & Frank, 2012) whereby decreased memory capacity might lead to a higher salience of more recent outcomes and therefore higher learning rates (Collins, Ciullo, Frank, & Badre, 2017). We also find support for this notion in our data, whereby poorer WM performance was strongly linked to higher belief volatility and higher choice uncertainty. However, despite sulpiride decreasing WM capacity in our cohort, including WM data in the model did not affect inference about sulpiride effect on belief volatility, nor on choice stochasticity. This increase in choice uncertainty or stochasticity under sulpiride that is therefore likely not due to failures in working memory capacity. Instead it could have been due to participants being less motivated to maximise outcome, and therefore less likely to behave as a rational “homo economicus” (Camerer, 2003). Were that the case in our study, one would expect a different behavioural pattern in the single shot trust games. Participants under sulpiride should

behave less as rational agents and therefore would be less likely to punish betrayals and reward trusting behaviour (Fehr, Fischbacher, & Gächter, 2002).”

2.) The justification for the inclusion of only males in the study warrants mention in (1.) the methods and (2.) as a limitation should be mentioned in the discussion. It must also be clear in the abstract. While it is true that historically females have been excluded from some studies involving dopaminergic modulation over concerns of variation across the ovarian cycle, more recent evidence (I assumed published before the data presented here was collected) has failed to find differences in D2 receptor availability across the ovarian cycle (Petersen et al., 2021) and other groups have taken steps to ensure equal consideration (e.g. studying at specific phase of the ovarian cycle). One purpose of this kind of caveat is to ensure that future replication efforts and related studies do consider enrollment of female research subjects.

A: The inclusion of males only is now mentioned in the abstract. It was mentioned in the methods already before. We agree that this is limitation that has become even more salient recently. We now discuss this at the end of the discussion on p.14:

“Several important limitations should be kept in mind when considering the generalizability of the findings in this study. First, the sample was limited to male participants. This restriction was initially motivated by the notion that including female participants would require more than doubling the sample size due to increased variance of dopamine availability across the ovarian cycle, however recent work shows there is no support for this (Petersen et al., 2021). Given that there are important sex differences in responses to antipsychotics, both in terms of efficacy and side-effect profiles (Hoekstra et al., 2021), future work should prioritize studies in females. Second, despite our hypothesis driven approach, the sample size of the genetic subgroups was small. Drug-gene interactions we observe should be interpreted with this in mind. Although we do find a main effect of sulpiride both on increased investment change as well as on belief volatility, this finding was not preregistered and is advised to be taken as preliminary.”

3.) The authors provide clinical relevance of their findings to psychosis treatment. One concern here is the extent to which comparisons can be made between mechanisms in healthy and clinical groups. For example, meta-analytic evidence indicates that the Taq1A polymorphism in schizophrenia does not affect responsiveness to antipsychotic pharmacotherapy (Zhang et al., 2010). This would seem at odds with the observation that some sulpiride-induced effects depend on Taq1A carrier status. Similar baseline differences in synthesis capacity have been observed in schizophrenia (Fusar-Poli & Meyer-Lindenberg, 2013; Howes et al., 2012). Recognition of these kinds of differences will be helpful when drawing inferences about psychopathology.

A: We acknowledge that the degree to which our findings can apply to clinical research is limited and have adapted this paragraph of the discussion accordingly (p.14):

“Our findings might be particularly relevant for understanding the effects of antipsychotic medication in patients with psychosis, a disorder characterized by rigid beliefs of persecution, underlined by a profound lack of trust in others (Freeman, 2016; Fuchs, 2015). Previous studies with repeated trust games showed that patients with psychosis have lower initial trust and find it hard to change their beliefs (Fett et al., 2012; Gromann et al., 2013). Neurocomputational accounts of delusions suggest that hyperactivity of D2 receptors in patients leads to increased precision beliefs that result in rigid convictions held with high confidence (Adams et al., 2013; Sterzer et al., 2018) and a recent paper shows that higher belief instability in patients with schizophrenia predicts responses to psychotherapeutic treatment (Hauke et al., 2022). This suggests that decreasing belief rigidity through D2 antagonism could be an important contributor to the success of adjunct psychosocial treatment. However, there are profound differences between effects of repeated use of antipsychotics in patients and

acute D2 antagonism in healthy volunteers. For example, rodent studies show that although in healthy animals D2 antagonism increases activity of midbrain dopaminergic cells, this pattern is reversed in an established animal model of schizophrenia (Valenti et al., 2011). Furthermore, despite rapid receptor blockade of D2 antagonists, the inhibition of excessive dopaminergic signalling, proposed to underly the therapeutic effects, comes only after weeks of treatment (Grace et al., 1997; Kapur et al., 2006). Our data also suggest that the therapeutic effect could be larger in patients that are A1 allele carriers of the Taq1a polymorphism. Yet there is no evidence for this (Zhang et al., 2010), despite higher dopamine synthesis being the most likely biomarker of psychotic symptoms (Fusar-Poli & Meyer-Lindenberg, 2013; Howes et al., 2012), and a predictor of response to antipsychotic treatment (Jauhar et al., 2019). A translation of our findings to clinical practice will require more work with targeted patient populations."

4.) There are a couple of places in the manuscript where an extra sentence or even a word or two could go a long way to aid interpretation of the results. For instance, "Importantly we observed that belief volatility ω correlates with reciprocity...and that the log transformed choice uncertainty parameter γ correlates negatively with the proportion of mistake trials...implying higher randomness in investment selection." Here there is a clear interpretation for the gamma but not omega parameter. Similarly, in the first paragraph of the section titled "Genotype-dependent effects of D2/D3 receptor antagonism on belief volatility and precision weights", there is no statement about the directionality of effects, just whether there are significant main or interaction effects on belief volatility. This contrasts the following paragraph where there is a clear statement ("participants under sulpiride have higher precision weights"). So, some clarity and consistency throughout the paper would go a long way.

A: Thank you for this comment and the concrete suggestions. We have replaced the title of the section with a title that is more consistent with titles of other sections. We have also tried to be more rigorous in explaining the implications of model checks throughout this section. For instance, the mentioned paragraph was adapted to (p.7):

"To determine how noisy trials are represented in the model, we defined mistake trials as trials where participants either decreased their investment after a positive back-transfer or increased their investment after a betrayal (for exact definition see Supplementary Note 1). Importantly, we observed that belief volatility ω correlates with reciprocity ($r = 0.277$, $t = 2.476$, $df = 74$, $p = 0.016$, Fig. 3d) confirming that higher trial-by-trial uncertainty of beliefs lead to a higher chance of reciprocal behaviour. The log transformed choice uncertainty parameter γ correlates negatively with the proportion of mistake trials ($r = -0.592$, $t = -6.3254$, $df = 74$, $p < 10e-3$, Fig. 3d) implying higher randomness in investment selection."

5.) Can the authors provide more interpretation on the significance of belief volatility being higher for the bad trustee? While the interactions observed for this effect are weaker, they might hold interesting value for understanding the specific conditions under which precision of priors depends on negative outcomes or even the relationship between trustworthiness and belief shifts.

A: We agree that the manuscript could profit from a more elaborate discussion on belief asymmetries across the good and the bad trustee, especially given the dichotomy of dopamine D1 and D2 receptors when processing positive and negative outcomes. We have added the following paragraph to the discussion (p.12):

"One fundamental distinction that separates risky decision-making under asocial compared to social conditions is aversion to betrayal (Fehr, 2009). People are less risk-taking in social interactions and might be particularly sensitive to indications of untrustworthy interaction partners (Bohnet et al., 2008). Using a similar model to ours, previous work has shown that belief volatility was higher when assessing (morally) bad agents (Siegel et al., 2018). In our study we also find higher belief volatility when

playing against the bad trustee. Although the drug effects on belief volatility were present across both trustees, the effects were stronger for the bad trustee. One reason for this could be that because participants initially expected higher trustworthiness and higher rates of positive back transfers, there was more to learn when playing against the bad trustee and therefore more variance across investment behaviour. This asymmetric increase in sensitivity to negative outcomes would also be in line with the notion that D1 and D2 receptors in the striatum contribute to positive and negative outcome processing, via the “Go” and “No-Go” pathways, respectively (Frank & O’Reilly, 2006; Frank et al., 2004). According to this circuit model, D2 antagonism mimics the dopaminergic dip that occurs in negative reinforcement and therefore enhanced learning to avoid an action with a negative outcome. However, contrary to what we observed, this model also predicts that blocking post-synaptic D2 receptors should decrease positive prediction error propagation.”

Minor Comments

1.) For what it is worth, there is additional work worth citing for the justification of using 800 mg of sulpiride. In clinical groups, at least, (Farde, 1988) show that in patients with schizophrenia, 800 mg (400 twice daily) corresponds to D2 occupancy between 68-82%.

A: Noted and done.

2.) Related to the general point about clarifying effects/directionality, the authors state “At this point we also note that there were no differences in initial beliefs (μ_0) about the trustworthiness” – it is unclear whether “differences” are referring to genotype or drug condition.

A: Actually, the observation that there are no differences in initial beliefs about trustworthiness across drug and genotype groups is an important point, and we now report both initial trustworthiness across the four groups as well as initial investments. Results, p.8, now read:

”At this point, we also note that on average, participants expected the trustee to reciprocate ($d = 0.717$, 95% CrI [0.406, 1.023], $P(d < 0) = 0$) with initial inferred probability of reciprocation being 0.67 (95% CrI [0.60, 0.73]). However, there were no differences between treatment groups in initial beliefs (μ_0) about the trustworthiness either overall ($d = -0.166$, 95% CrI [-0.697, 0.37], $P(b > 0) = 0.272$, Supplementary Fig. 4), in the A1+ group ($d = 0.111$, 95% CrI [-0.585, 0.802], $P(d < 0) = 0.383$), or in the A1- group ($d = -0.441$, 95% CrI [-1.164, 0.287], $P(d > 0) = 0.112$).”

3.) Supplementary Figure 4 needs some kind of detailed caption like the rest of the figures.

A: Noted and done.

4.) Potential typo: In the statement “sulpiride led to higher reciprocal behaviour (increased investment after negative back-transfer and decreased investment after positive back-transfer)” should it be increased after positive and decreased after negative?

A: This has been corrected – we thank the reviewer for spotting this mistake!

5.) It is not clear what the y-axis labeling of top two plots in figure 2B are representing. The image itself is unclear but a parenthetical note or something of that nature in the caption could help.

A: We agree, and the new version of the figure is hopefully clearer and better annotated.

6.) Were dependent variables z-scored within subjects or globally across all the data?

A: Across all the data. This information is added in the Methods:

“Throughout the behavioural analysis, we z-scored the dependent variables (across the whole group), coded the Treatment variable as 0 (placebo) and 1 (sulpiride), Back-transfer as 1 (equalize) and -1 (betray), Genotype as 0 (A1-) and 1 (A1+) and centred the trustee variable (0.5 for good, and -0.5 for bad trustee).”

7.) Potential typo: In the “Parameter Estimation” section of the methods, it reads 3200 iterations but from the code and description in the “Behavioural Analysis” it should be 3000, I think?

A: We agree. This has now been corrected.

8.) Typo in Fig 4a caption: “driven pa the A1- group.”

A: The previous Figure 4 (now Figure 5) described the results of sulpiride on choice uncertainty, where the effect was driven by the A1- group.

Overall, Mikus and colleagues present an impressive neurocomputational account of dopaminergic modulation of belief updating in a social context. This work will be of interest to social/cognitive psychologists, neuroscientists, and clinical investigators and provides an important contribution to the study of decision making.

A: We thank the reviewer for their kind words!

References

Farde, L. (1988). Central D2-Dopamine Receptor Occupancy in Schizophrenic Patients Treated With Antipsychotic Drugs. *Archives of General Psychiatry*, 45(1), 71. <https://doi.org/10.1001/archpsyc.1988.01800250087012>

Fusar-Poli, P., & Meyer-Lindenberg, A. (2013). Striatal Presynaptic Dopamine in Schizophrenia, Part II: Meta-Analysis of [18F/11C]-DOPA PET Studies. *Schizophrenia Bulletin*, 39(1), 33–42.

Howes, O. D., Kambeitz, J., Kim, E., Stahl, D., Slifstein, M., Abi-Dargham, A., & Kapur, S. (2012). The Nature of Dopamine Dysfunction in Schizophrenia and What This Means for Treatment: Meta-analysis of Imaging Studies. *Archives of General Psychiatry*, 69(8). <https://doi.org/10.1001/archgenpsychiatry.2012.169>

Petersen, N., Rapkin, A. J., Okita, K., Kinney, K. R., Mizuno, T., Mandelkern, M. A., & London, E. D. (2021). Striatal dopamine D2-type receptor availability and peripheral 17 β -estradiol. *Molecular Psychiatry*, 26(6), 2038–2047. <https://doi.org/10.1038/s41380-020-01000-1>

Zhang, J.-P., Lencz, T., & Malhotra, A. K. (2010). D 2 Receptor Genetic Variation and Clinical Response to Antipsychotic Drug Treatment: A Meta-Analysis. *American Journal of Psychiatry*, 167(7), 763–772. <https://doi.org/10.1176/appi.ajp.2009.09040598>

REVIEWERS' COMMENTS

Reviewer #1 (Remarks to the Author):

The authors have considerably revised their manuscript and ran additional analyses to address my concerns. They have clarified various important aspects of the study, including the basis for the directional hypothesis, the construct under study (notably the key difference between belief flexibility and volatility of beliefs as a function of the environment e.g., changes of mind), the design (which clearly controlled for changes of mind) and the (comparison) models. They also report additional analyses showing a striking association between working memory performance and belief volatility, while critically also ruling out a working memory mechanism the key effects of sulpiride on belief volatility. I have no further questions.

Reviewer #2 (Remarks to the Author):

The authors have successfully addressed my concerns and I am satisfied with their revision.

Reviewer #3 (Remarks to the Author):

I believe that the authors have done an excellent job of addressing all of my concerns. I appreciate the examination of the contributions of working memory and the manuscript is now stronger with more thorough reporting of potential mechanisms, implications, and elaboration on the computational framework.